



# Megacity and local contributions to regional air pollution: An aircraft case study over London

Kirsti Ashworth[1*], Silvia Bucci[2*], Peter J. Gallimore[3*], Junghwa Lee[4*,a], Beth S. Nelson[5*], Alberto Sanchez Marroquín[6*], Marina B. Schimpf[7*,b], Paul D. Smith[8], Will S. Drysdale[5,c], Jim R. Hopkins[5,c], James D. Lee[5,c], Joe R.Pitt[9], Piero Di Carlo[10†], Radovan Krejci[11†], James B. McQuaid[6†]

[1]Lancaster Environment Centre, Lancaster University, LA1 4YQ, UK
[2]Laboratoire de Météorologie Dynamique, UMR8539, IPSL, CNRS/PSL-ENS/Sorbonne Université/École Polytechnique, Paris, France
[3]Department of Chemistry, University of Cambridge, Lensfield Road, Cambridge, CB2 1EW, UK
[4]Institute of Meteorology and Climatology, Leibniz Universität Hannover, Hannover, Germany
[5]Wolfson Atmospheric Chemistry Laboratory, Department of Chemistry, University of York, YO10 5DD, UK
[6]School of Earth and Environment, University of Leeds, Woodhouse Lane, Leeds, LS2 9JT, UK
[7]Facility for Airborne Atmospheric Measurements (FAAM), Cranfield, MK43 0AL, UK
[8]Swedish University of Agricultural Sciences, Svartberget Fieldstation, SE-92291 Vindeln, Sweden
[9]School of Earth and Environmental Sciences, University of Manchester, Oxford Road, Manchester, M13 9PL, UK
[10]Department of Psychological, Health & Territorial Sciences, University "G. d'Annunzio" of Chieti-Pescara, Chieti – Italy
[11]Department of Environmental Science and Analytical Chemistry (ACES) & Bolin Centre for Climate Research, Stockholm University, S-10691 Stockholm, Sweden
[*]Participant of STANCO training school
[†]Co-PI of STANCO training school
[a]now at: Department of Modelling of Atmospheric Processes, Leibniz Institute for Tropospheric Research (TROPOS), Leipzig, Germany
[b]now at: German Aerospace Centre (DLR), Institute of Atmospheric Physics, Oberpfaffenhofen, Germany
[c]National Centre for Atmospheric Science, University of York, York, YO10 5DD

*Correspondence to*: Kirsti Ashworth (k.s.ashworth1@lancaster.co.uk)

**Abstract.** In July 2017 three research flights circumnavigating the megacity of London were conducted as a part of the STANCO training school for students and early career researchers organised by EUFAR (European Facility for Aircraft Research). Measurements were made from the UK's Facility for Airborne Atmospheric Measurements (FAAM) BAe-146-301 Atmospheric Research Aircraft with the aim to sample, characterise and quantify the impact of megacity outflow pollution on air quality in the surrounding region. Conditions were extremely favourable for airborne measurements and all three flights were able to observe clear pollution events along the flight path. A small change in wind direction provided sufficiently different airmass origins over the two days such that a distinct pollution plume from London, attributable marine emissions and a double-peaked dispersed area of pollution resulting from a combination of local and transported emissions were measured. We were able to analyse the effect of London emissions on air quality in the wider region and the extent to which local sources contribute to pollution events.

The background air upwind of London was relatively clean during both days; concentrations of CO were 88-95 ppbv, total (measured) volatile organic compounds (VOCs) were 1.6-1.8 ppbv, and $NO_x$ were 0.7-0.8 ppbv. Downwind of London, we encountered elevations in all species with CO >100 ppbv, VOCs 2.8-3.8 ppbv, $CH_4$ >2080 ppbv and $NO_x$ >4 ppbv, and peak concentrations in individual pollution events higher still. Levels of $O_3$ were inversely correlated with $NO_x$ during the first flight, with $O_3$ concentrations of 37 ppbv upwind falling to ~26 ppbv in the well-defined London plume. Mass balance techniques were applied to estimate pollutant fluxes from London. Our calculated $CO_2$ fluxes are within 10% of those estimated previously, but there was a greater disparity in our estimates of $CH_4$ and CO.

On the second day, winds were lighter and downwind $O_3$ concentrations were elevated to ~39-43 ppbv (from ~32-35 ppbv upwind), reflecting the contribution of more aged pollution to the regional background. Elevations in pollutant concentrations were dispersed over a wider area than the first day, although we also encountered a number of clear spikes from local sources.





This series of flights demonstrated that megacity outflow, local fresh emissions and more distant UK sources of pollution all contribute to pollution events in the southeast of the UK. These sources must therefore all be well-characterised and

constrained to understand air quality around London.

## 1 Introduction

Over half of the world's population live in urban areas, a figure expected to rise to ~70% by 2050. There are currently 37 megacities (cities with population >10 million), mostly in South and East Asia, and this number is rapidly increasing with a further 6 likely to reach this size by 2030. The speed of urban growth is such that megacities act as large pollutant sources

that strongly influence the environment of the surrounding region.

More than 4 million deaths each year are attributed to ambient air pollution, with >90% of the urban population exposed to air pollution levels that exceed World Health Organisation (WHO) limits (WHO, 2018). In the UK, urban air quality is an issue of increasing public concern with air pollution in London a particular focus. Measurements at Marylebone Road recorded an annual average concentration of 44 ppbv of $NO_2$ in 2017 (over twice the European Environment Agency's limit)

with 38 exceedances of the hourly limit (down from 122 in 2012) and 12 exceedances of the daily maximum $PM_{10}$ limit of 50 µg m$^{-3}$ (down from 48 in 2012; WCC, 2018).

London has been the target of numerous ground-based and airborne measurement campaigns attempting to understand the sources, formation and extent of air pollution in the city and across the wider region. The most relevant of these to the current study include RONOCO (Role of Nighttime chemistry in controlling the Oxidising Capacity of the atmOsphere) in

2010-11 (Stone et al., 2014), EM25 (Emissions around the M25) campaign in 2009 (McMeeking et al., 2012), ClearFLo (Clean air for London) in 2012 (O'Shea et al., 2014), flights off the southern and eastern coasts of the UK during EUCAARI-LONGREX in 2008, (e.g. Hamburger et al., 2011, Highwood et al., 2012), and innovative sorties to calculate emission fluxes (Shaw et al., 2015). Synoptic conditions, wind speed and direction were highly variable during these campaigns, resulting in large ranges of measured trace gas and particle concentrations.

The flight paths during the EM25 campaign (McMeeking et al. (2012) and one daytime flight undertaken during RONOCO (Aruffo et al., 2014) were similar to ours, circuiting London above the M25 and overflying the southern and eastern coast of the UK. However, Aruffo et al. (2014) reported very weak north-easterly winds similar to one of the EM25 flights but in contrast to the west and south-westerly observed during our three flights. The other EM25 flights encountered clear westerly and easterly air flows of different strengths making interpretation and apportionment difficult. Concentrations of most trace

gases measured by Aruffo et al. (2014) were low with average levels of $NO_x$ <2 ppbv and ozone ~40 ppbv throughout the flight. However, on each of the three circuits around the M25 orbital motorway, a clear plume of pollution from Greater London was sampled to the west. In the plume $NO_x$ levels were enhanced by as much as 27 ppbv resulting in substantial titration of ozone which reduced $O_3$ concentrations to as low as 16 ppbv. This effect peaked over the city of Reading (population >300,000) where it is likely that local emissions enhanced the plume. While CO concentrations were also

elevated within the plumes, strong peaks were also observed to the east of London presumably as the result of large local point sources.

These observations match those of the EM25 campaign. McMeeking et al. (2012) also report substantial elevations in $NO_x$ and CO in the London pollution plumes along with clear evidence of ozone titration. Aerosol mass concentrations were also enhanced in the plumes (~10 µg m$^{-3}$, compared with ~6 µg m$^{-3}$ upwind of London). During their flight B460, when the wind

was also easterly, the peak of the plume was again encountered over Reading.

O'Shea et al. (2014) demonstrated the potential of using aircraft measurements to perform a pollutant mass balance for the Greater London area. Such an approach can serve as an independent verification and constraint of bottom-up emission





inventories under meteorological conditions that ensure a clear well-defined spatially-constrained plume downwind of an urban source area with relatively homogeneous clean air upwind. During one flight in July 2012 with suitable meteorology, the authors report enhancements of ~3% in $CO_2$, ~4% in $CH_4$ and ~31% in CO relative to the mean background concentration (i.e. that observed upwind of London). The authors used the observed increases to back-calculate an emission flux for Greater London and compared their estimates to the total emissions of $CO_2$, $CH_4$ and CO from London in the National Atmospheric Emissions Inventory (NAEI). Airborne estimated fluxes were found to be a factor of 2.3, 3.4 and 2.2 higher for $CO_2$, $CH_4$ and CO than the NAEI dataset. However, as the authors point out, NAEI values are annual while the airborne measurements are for a single day; this temporal difference is likely contributing at least in part to the discrepancy, highlighting one difficulty in interpreting and evaluating aircraft atmospheric measurement data.

Shaw et al. (2015) report mixing ratios of anthropogenic VOCs, $NO_x$ and $O_3$ measured from the Natural Environment Research Council's (NERC's) Dornier 225 aircraft from six flights carried out in June-July 2013. Mean concentrations of benzene, toluene and $NO_x$ were highest over Inner London (0.20±0.05, 0.28±0.07 and 34.3±15.2 ppbv respectively) and peaked during morning rush-hour, when clear evidence of $O_3$ titration was also observed. Mixing ratios were generally lower over Greater London and the surrounding suburbs although elevated $NO_x$ levels were encountered in the outflow from London Heathrow airport consistent with aircraft and road traffic emissions.

Here we report on a series of three flights conducted on 3rd-4th July 2017 during STANCO (School and Training on Aircraft New Techniques for Atmospheric Composition Observation), organised on behalf of EUFAR (European Facility for Aircraft Research). Each flight circled London to detect and sample the urban plume, and to determine the relative contributions of London outflow and local sources to other pollution plumes measured during the flights.

The next section provides a short overview of the three flights, the on-board instrumentation, the sampling conducted and the back-trajectory analysis performed. We present our results in Section 3, and analyse the observations in more detail. We discuss the sources for specific pollution events that we observed during each flight and conclude with a brief summary in Section 4.

## 2. Methods

### 2.1 Overview

Full details of the flight paths are given in Table 1 and Fig. 1. Flight C016 took off from Cranfield airfield at ~11:10 on 3rd July and flew clockwise around London; flights C017 and C018 departed at ~09:40 and 14:20 respectively on 4th July, flying counter-clockwise due to a shift in wind direction overnight. In all three cases conditions were settled with relatively good visibility. Cruising altitude was 800-1000 m, based on the on-board GPS-inertial navigation system, dropping to ~150 m over land and 25 m over the sea to sample specific plumes.

The dates and times of the three flights are listed in Table 1. Fig. 1 shows the flight pattern of the flights which were designed to intercept and sample the pollution outflow from London and probe local pollution across SE England.

Due to the change in synoptic situation between the two flight days we observed two very different patterns of pollution, both local sources and the emission outflow from London. Consistent westerly winds on 3rd July gave rise to a distinct "plume" east of London over the Thames Estuary, with elevated gas and particle concentrations relative to the upwind air west of London. The clear definition of the plume edges allowed us to quantify the outflow of pollution from London using a mass balance approach (see Section 3.3.1). Relatively stagnant conditions and the shift in wind direction on 4th July reduced the influence of London emissions on the surrounding region. High pollutant levels measured during flights C017 and C018 could thus be attributed to local sources.



### 2.2 Sampling platform

The UK's Facility for Airborne Atmospheric Measurements (FAAM) BAe-146-301 Atmospheric Research Aircraft (hereafter "FAAM BAe-146") operated by the UK's provided the airborne science platform. The aircraft has a working altitude range of 100 to 30 000 feet (Petersen and Renfrew, 2009) and a core instrument payload that has been described in full elsewhere (e.g. Harris et al., 2017). The instruments relevant to the current series of flights are described below.

#### 2.2.1 Meteorological measurements

Temperature, wind vector, pressure and humidity are all core measurements. Temperature was recorded with an accuracy of ±0.3 K using Rosemount (Rosemount Aerospace Ltd., UK) type 102 de-iced (Rosemount 102BL) and non-de-iced (Rosemount 102AL) Total Air Temperature sensors (Petersen and Renfrew, 2009; Harris et al., 2017). Pressure and 3-D wind vectors were recorded with estimated uncertainties of 0.3 hPa and 0.2 ms$^{-1}$ respectively (O'Shea 2014; Allen et al., 2011). Humidity was measured only in cloud-free air with a General Eastern 1011B chilled mirror hygrometer. Altitude, position and aircraft velocity data were recorded at 32 Hz by a GPS-aided Inertial Navigation system. The measurement protocol for these and other atmospheric parameters has been described in detail by Petersen and Renfrew (2009) and Allen et al. (2011).

#### 2.2.2 Trace Gas Concentrations

Volatile Organic Compounds (VOCs) were sampled using the whole air sampling (WAS) system fitted to the rear-hold of the aircraft. The system consists of sixty-four silica passivated stainless steel canisters (Thames Restek, Saunderton UK) connected via a 3/8 inch diameter stainless steel sample line to an all-stainless steel assembly metal bellows pump (Senior Aerospace, USA) which draws air from the port-side sampling manifold and pressurised air into 3 L canisters to a maximum pressure of 3.25 bar (giving a useable analysis volume of up to 9 L). The collection time of ~20s equates to a smoothed average VOC concentration over ~2 km (Lee et al., 2018). The WAS canisters were analysed by withdrawing and drying 700 ml samples of air using a glass condensation finger held at -40 °C. These samples were preconcentrated using a Markes Unity2 pre-concentrator (fitted with an ozone precursors adsorbent trap) and CIA Advantage autosampler (Markes International Ltd), and then transferred to the GC oven for analysis as described by Hopkins et al. (2011). Further details are given by Lewis et al. (2013) and Lidster et al. (2014).

In-situ measurements of NO were made using a custom built chemiluminescence instrument (Air Quality Design Inc) with NO$_2$ measured by photolytic conversion to NO on a second channel. In-flight calibrations were carried out above the boundary layer at the beginning and end of each flight by adding a small flow of 5ppmv NO in nitrogen (BOC) to the sample inlet. The NO$_2$ conversion efficiency was measured using gas-phase titration of the NO by O$_3$ in the calibration to NO$_2$. The calibration factors were interpolated throughout the flight to account for any sensitivity drifts in the instrument. Detection limits are ~22 pptv for NO and ~23 pptv for NO$_2$ for 1 Hz averaged data, with estimated accuracies of 15% for NO at 0.1 ppbv and 20% for NO$_2$ at 0.1 ppbv

Continuous 1 Hz measurements of CO$_2$ and CH$_4$ were made by Fast Greenhouse Gas Analyser (FGGA; Model RMT-200, Los Gatos Research, USA). The instrument was calibrated ~hourly using a two-point calibration by sampling two cylinders of air containing CO$_2$ and CH$_4$ at mole fractions that span the normal measurement range. A third "target" cylinder containing intermediate mole fractions of CO$_2$ and CH$_4$ was sampled approximately mid-way between hourly calibrations to allow for an assessment of the calibrated data quality. During 12 flights conducted between May-July 2017, the average difference between the target cylinder measurements and the known cylinder composition was -0.047 ppmv for CO$_2$ and -0.49 ppbv for CH$_4$. The standard deviation of this difference at 1 Hz was 0.348 ppmv and 1.64 ppbv, respectively. Combining these with the uncertainties associated with water vapour correction (0.150 ppmv and 1.03 ppbv, respectively)



and the certification of the target cylinder (0.075 ppmv and 0.76 ppbv, respectively) yields nominal total uncertainties of 0.386 ppmv for $CO_2$ and 2.08 ppbv for $CH_4$ at 1 Hz. A detailed description of the in-flight calibration system is given by O'Shea et al. (2013).

Measurements of CO were made with a fast-response vacuum-UV resonance fluorescence spectrometer with an uncertainty of 2% (Model AL5002, Aerolaser GmbH, Germany; Gerbig et al. 1999). Ozone ($O_3$) concentrations were measured using a UV photometric analyzer (Model TEi-49i, Thermo Fisher Scientific Inc., USA).

**2.2.3 Aerosols**

Sub-micron non-refractory aerosol composition was measured by an Aerodyne Research (Billerica, MA, USA) Compact

Time of Flight (CTOF) type AMS (Canagaratna et al., 2007; Drewnick et al., 2005). The sampling strategy has been described in previous studies (Crosier

et al., 2007; Capes et al., 2008; Morgan et al., 2009). The measurement accuracy is estimated to be 10 % (not considering the collection efficiency uncertainty) with detection limits for organics and ammonium ~40 ng m$^{-3}$, and for nitrate and sulphate ~5 ng m$^{-3}$ (Drewnick et al., 2005). Ionisation efficiency of nitrate and relative ionisation efficiencies of ammonium and

sulphate were obtained from calibrations performed using monodisperse ammonium nitrate and ammonium sulphate (see Robinson et al., 2011 and Morgan et al., 2010b). The temporal stability of total aerosol was monitored using a condensation particle counter (CPC; Model 3786, TSI Incorporated, MN, USA) at 1 Hz. An optical particle counter (OPC; Grimm Aerosol Technik GmbH & Co. KG, Germany) was used to correctly count and size aerosol particles (Allen et al., 2011). Aerosol scattering at 450, 550 and 700 nm was recorded using a three-channel TSI 3563 Integrating Nephelometer.

**2.3 Air mass transport**

We make use of the FLEXPART Lagrangian particle dispersion model (Stohl et al., 2005, and references therein) adapted for WRF (Brioude et al 2013) to characterize air mass transport conditions during the STANCO campaign. Meteorological input from WRF is provided with hourly time step at a spatial resolution of 3km x 3km. Clusters of 500 back-trajectories are computed back in time for 24 hours with a 1-hour time step.

The output is a gridded "footprint emissions sensitivity" of the retroplume (as described in Stohl et al., 2007). It quantifies the residence time of the back trajectory plume over each bin and, hence, the potential contribution of such bin to the air mass composition at the point of the trajectories' release. When looking for correspondences with ground emissions, we select the back-trajectories from below the boundary layer, as interpolated by FLEXPART from the WRF simulations.

**3. Results**

**3.1 Meteorology and air mass history**

Meteorological conditions on 3$^{rd}$-4$^{th}$ July 2017 are summarised in Fig. 2 which shows the low-level (850 hPa) wind fields with geopotential height and Liquid Water Path (LWP) during the flight period from the ERA-interim ECMWF re-analysis data.

During C016 the mean flow at 850 hPa was mainly westerly with winds <15m s$^{-1}$ across the London area, giving favourable

conditions to study the London plume (see Section 3.3.1 for further details). There were clouds and slight precipitation in the southwest flight quadrant, and sun in the east. The SkewT-logP diagrams show that the lifted condensation level was ~890 hPa, effectively constraining near-surface emissions below this height. Fig.3a shows the height of the mixed layer varied between ~800m (during flight C016 on 3$^{rd}$ July) and ~1500m (during C018, the afternoon flight on 4$^{th}$ July). Our airborne observations show good agreement of mixing layer height with those obtained from a radiosonde ascent over nearby

Nottingham at 00:00 UTC on 04$^{th}$ July (Fig. 3b); our sounding profiles show that all sampling was performed within the



mixed layer.

The high-pressure system that brought westerly flow on 3$^{rd}$ July moved to the north overnight, bringing south-westerlies for both flights on 4$^{th}$ July. Windspeed also dropped to <10 m s$^{-1}$ and urban air pollution was dispersed rather than concentrated into a plume. The local air pollution sources observed during flights C017 and C018 are further discussed in Sections 3.3 and 3.4.

### 3.2 Airborne observations

Fig. 1 shows the path of FAAM BAe-146 during each of the three flights, broken into segments of equal duration, which are numbered to enable us to locate the observed features geographically. Time series of the aircraft altitude and continuously measured gas-phase concentrations and aerosol number density are plotted for the three flight paths in Figs. 5-7; the numbers shown in the upper panel of each correspond to the numbered flight segments in Fig. 1. In addition to the suite of real-time continuous measurements sampled from the aircraft 24, 14, and 24 WAS were collected during each of the flights and later analysed for VOC concentrations. Table 2 shows average concentrations of all trace gases and aerosol number density for the whole flight, upwind (clean) flight segments, and downwind (plume) flight legs at the nearest altitude to the upwind segment. It should be noted that as WAS were manually initiated, in response to observed elevations in other trace gases as well as to target flight segments up- and down-wind of London, so the data must be considered skewed to more polluted locations. A further caveat is the very small sample size.

Table 2 shows the clear enhancement in gas-phase concentrations downwind of London. CO concentrations are as low as ~88-95 ppbv on the upwind flight segments but increase by ~10 ppbv in the plume on each flight. Enhancements of CH$_4$ are around 20% in all downwind plumes (rising from ~2.01-2.05 to ~2.05-2.08 ppmv). NO$_x$ reached peaks of >14 ppbv on 3$^{rd}$ July and >4.5 ppbv on 4$^{th}$ downwind of London compared to levels between ~0.7-0.8 ppbv in the relatively clean upwind air. NO$_x$ concentrations were highly variable across all 3 flights as expected for such short-lived species associated with fresh local emissions. Total VOC concentrations rose by a factor of ~2 (from ~1.6-1.8 to ~2.7-3.7 ppbv) although the changes in individual species varied between the flights. The only exception to this pattern are ozone levels during C016 which are considerably lower in the plume (~26 ppbv) than along the upwind flight segment (36.5 ppbv). Interestingly, all three flights had similar concentrations of O$_3$ upwind of London (~32.6-36.5 ppbv). High O$_3$:NO$_x$ ratios are characteristic of aged air masses and suggests that the pollution encountered along the upwind flight segments to the west (C016) and south-west (C017 and C018) of London is the result of transported rather than local fresh emissions. This is discussed in further detail in Section 3.3.1.

The other striking difference between the flights, also symptomatic of the origin of the transported air, is the aerosol number density. Both C016 and C018 encountered much higher numbers upwind than in the London outflow (2x10$^4$ and 1.5x10$^4$ cm$^{-3}$ vs. 7x10$^3$ and 5x10$^3$ cm$^{-3}$ respectively); in both cases, this background air had travelled from the west to south-west. By contrast, flight C017 sampled air transported from the west to north-west of the UK and aerosol number density was lower upwind of London (2.5x10$^3$ vs. 7x10$^3$ cm$^{-3}$ downwind), suggesting the enhancement was due to a strong source SW of London rather than local to the flight track.

### 3.2.1 Flight C016: Westerly advection

A large part of C016 took place east of the UK coast, flying mostly below 800 m altitude over the sea, where we sampled air inside the PBL in conditions of high RH (values between 90 and 100%) and a potential temperature of ~290K. Pollutant levels during this flight were higher than the two later (inland) flights. The enhancement in the trace gas concentrations and aerosol number density can be almost entirely attributed to pollutants emitted and advected from the UK with little influence of continental Europe. Air mass back trajectories for the flight segments to the east of London are shown in Fig. 5. The sharp edges to the plume can be deduced from these snapshots in time, with the air masses intercepted at 11:48:00 and 11:55:00





traversing London but those at 11:41:00 and 12:01:00 bypassing the city and bringing cleaner air from other regions. In this downwind section of the flight (2-9 of Fig. 6 and first panel of Fig. 4) CO concentrations ranged from 90-120 ppbv. We also observed the highest values of $NO_x$, often in excess of 10 ppbv and peaking at 14.6 ppbv, and elevated concentrations up to

450 ppmv of $CO_2$ and up to 2 ppbv of $CH_4$. Aerosol number density was mostly $<10^4$ $cm^{-3}$, with the exception of two layers between 600 and 700 m altitude where numbers peaked to $3 \times 10^4$ $cm^{-3}$ east of and parallel to London (segments 6-7). Above the mixed layer and at higher altitudes >1500m we did not observed any striking feature.

**3.2.2 Flights C017-C018: South-westerly advection**

Meteorological conditions were more quiescent on Tuesday 4th July with relatively slack air flow from WSW to WNW

throughout the day, giving way to some localised re-circulation, particularly to the northeast of London (the origin and transport of air masses are discussed in more detail in Section 3.3.2). We did not encounter a clear London plume, but instead were able to identify other more local pollution events which are presented in Sections 3.3.2 and 3.4.

Flights C017 and C018 followed the same flight plan, circumnavigating London in a clockwise direction along the same route and altitudes as far as possible, as shown in Figs. 1 and 6-7. The initial altitude was 1500 m during both flights on 4th

of July, before a descent to 700 m to the west and south of London and then to 25 m over the Dover Straits and English Channel (flight segments 4-5 on Figs. 6 and 7) where we were able to sample distinct plumes from marine traffic (see Section 3.4). There then followed the series of reciprocal runs over East Anglia (segments 6-11) where a diffuse plume of pollution was encountered with elevated CO and $CH_4$ concentrations and, to a lesser extent, aerosol number density over a relatively large area. Within this, two distinct plumes of pollution were observed and sampled in both flights - an interesting

case of transport from two distinct outflow plumes which is analysed in more detail in Section 3.3.

The humidity and temperature during these flights were similar to those during C016, with RH varying between 95% and 100% and potential temperatures between 290 and 295K. However, conditions during C017 and C018 differed in several notable ways. The morning flight (C017) was characterised by relatively stagnant winds (see Fig. 2) and a low mixed layer depth (~800m). Pollutant concentrations were the lowest sampled (Fig. 6). During the afternoon, wind speed increased and

the height of the PBL rose to ~1500m, resulting in important changes in the observed air composition (Figs. 6 and 7). Notably, aerosol number density reached $2 \times 10^4$ $cm^{-3}$ between flight segments 2 and 4, apparently associated with an air mass originating from SW England. This is in sharp contrast to the morning flight ($<5 \times 10^3$ $cm^{-3}$ in this area) with the difference likely caused by the higher afternoon boundary layer uplifting local particles from southern England as no enhancement was observed during the latter stages of the flight when the air masses were transported from more northern and central regions.

During C017, CO concentrations were <100 ppbv, with the exception of a peak reaching 115 ppbv, associated with an enhancement in $NO_x$ of up to 2 ppbv, detected at an altitude of around 500m in the vicinity of Gatwick airport (at 51ºN, 0.55ºE; close to segment 4 on Figs. 1 and 6). This feature was not observed on flight C018 (Fig. 7). Fig. 8 shows air mass footprints from FLEXPART back- trajectories for the 5-minute time interval during which the plume was observed on-board C017 and the equivalent interval for the afternoon flight, and indicates the difference is the result of a greater influence of

transported pollution from land-based sources in the morning, with the sampled air spending more time over the sea in the afternoon. The NAEI emission inventory suggests this was likely local pollution from the Brighton area and A26 major road.

Aside from these distinct events, similar patterns were observed in pollutant concentrations and aerosol number density although the absolute values differed between the two flights. In particular, high concentrations of CO, $CH_4$, and $NO_x$ were measured in the NE quadrant of both flights over northern East Anglia (around 1.5 ˚E, 52.5 ˚N, see Figs. 1, 6 and 7) and

reciprocal runs were performed above this location, to sample the pollution at multiple heights in the boundary layer. CO reached values of 120 ppbv, $NO_x$ ~5 ppbv, $O_3$ concentrations >50 ppbv (compared with <40 ppbv during the first part of C018 and throughout the other flights) and $CH_4$ >2 ppbv. $CO_2$ levels however were always <420 ppmv. Back trajectories, considered alongside NAEI emission sources suggest this was associated with transport from a wider region including Wales


and NW England over the previous 24 hours, which had then been advected northward in the final 6 hours to reach the
Norwich region. Away from this location, little pollution was encountered during either flight with $NO_x$ generally <2 ppbv,
$CH_4$<1.95 ppbv, $CO_2$ <430 ppmv and aerosol number density <$10^4$ cm$^{-3}$.

**3.3 Megacity outflow vs local sources**

Each of the three flights followed similar flight paths, circling London just beyond the outer ring road (M25), in order to
intersect and sample relatively clean "background" air upwind of the city (segments on C016, C017 and C018 respectively)
and polluted outflow downwind. Clear increases in pollutant concentrations were detected in the urban plume. Small
differences in windspeed and direction across the three flights resulted in air masses with very different origins contributing
to the background composition and to individual pollution events. We were thus able not only to analyse the effect of
London emissions on air quality in the wider region but also the extent to which local sources contributed significantly to
specific observed pollution events.

**3.3.1 London plume: Flight C016, 3$^{rd}$ July**

A narrow well-defined plume of pollution was encountered downwind of London (flight segments 3-9 in Figs. 1 and 5). A
series of reciprocal runs was performed in this outflow over the Thames Estuary at altitudes between 100 and 800 m
capturing its vertical profile. In addition to the continuous measurements, 13 WAS were collected during these flight legs.
Table 2 shows the average concentrations of gas-phase pollutants and aerosol number density across segments 4, 6 and 7
(average altitude ~450 m). Flight segment 12 in Figs. 1 and 5 lies directly upwind of the city and provided a contrasting
relatively clean air mass (as evident in Table 2). Five WAS were made along this leg at an average altitude of 550 m.

The outflow from London is easily identified by the sharp spikes in $NO_x$, CO, $CO_2$ and $CH_4$ concentrations seen in segments
3-9 in Fig. 5, which anticorrelate with $O_3$ concentrations. $O_3$ concentrations decrease sharply (to ~22-25 ppbv) in the plume
due to NO titration and are highest (~35-40 ppbv) in the upwind air mass due to the formation of $O_3$ and other secondary
pollutants from photochemical ageing of more distant emission sources. Total (measured) VOC concentration was also
elevated in the plume (peaking at 5.9 ppbv) compared with upwind air (max 1.8 ppbv). However, proportions of longer-lived
compounds (e.g. ethane and propane) were higher upwind (~0.5 *vs.* ~0.3 and 0.35 *vs.* 0.17 ppbv). The ratios of benzene to
toluene (B:T; 1.78 *vs.* 0.63) and $O_3$:$NO_x$ (49.6 *vs.* 8.4) being higher upwind than in the plume further reinforce that the
upwind airmass is more aged and are typical of urban plumes (McMeeking et al., 2012). This is also evident in the ratios of
benzene to acetylene, ~0.7 upwind and ~0.4 downwind, although our value of 0.4 is slightly higher than has been previously
reported for London (Parrish et al., 2009; McMeeking et al., 2012; von Schneidemesser et al., 2010).

Fig. 9 shows the concentrations of key VOCs for each reciprocal run in the plume; the altitude of each is indicated on the x-
axis. The highest absolute concentrations occurred at altitudes between ~200-600 m. This is suggestive of pollution being
lofted above a layer of cooler surface air outside of the urban heat island, i.e. the urban boundary layer phenomena. Overall,
our observations support the conclusion that it was London outflow that we sampled during the reciprocal runs over the
Thames Estuary, with little evidence of strong contributions from local emission sources. The relatively strong (>15 m s$^{-1}$)
prevailing south-westerly ensured measurements from FAAM BAe-146 provided a data footprint large enough to allow the
calculation of regional-scale CO and $CH_4$ fluxes from the plume using a mass balance approach. While measurements only
cover a small percentage of the vertical profile, the high sampling rate and spatial distribution of the data in the plume
allowed for the interpolation of the data for the flux to be calculated using this method. Good characterization of the
background air, found by measurements around the edges of the London plume, was also integral for this analysis.

Several secondary plumes from shipping emissions were removed from the dataset before mass balance analysis was
performed. Discrete data points were then interpolated onto a 19x19 grid consisting of 8412 m by 38 m grid boxes in the


horizontal and vertical respectively. Kriging was achieved using the MATLAB "EasyKrig3.0" program (Chu, 2004). A

vertical plane for the downwind plume was produced, along with the wind vector perpendicular to these planes, using the

methodology described by Kitanidis (1997) and May et al. (2009). Vertical background runs were created by linearly

interpolating between the northern-most and southern-most data outside of the plume for each run. These were then

interpolated using kriging to produce corresponding 19x19 grid boxes for the background planes.

Pressure and temperature were accounted for in this analysis using the pressure and temperature dependent conversion of

species concentrations from ppbv to mg m$^{-3}$. The total flux can thus be calculated for species X, where X is CO, $CO_2$ or $CH_4$,

using Equation 1.

$$Flux = \int_0^z \int_A^B (S_{ij} - S_0) . U_{\perp ij} \, dx \, dz \qquad (1)$$

where $S_{ij}$ is the mole fraction of species S for coordinates in the downwind vertical plane, AB and $S_0$ is the background

vertical plane, and $U_{\perp ij}$ is the vertical plane of the wind vector perpendicular to the aircraft. The flux is then integrated for

altitude ($z$) of 0 m to the top of the plume at ~900 m.  The downwind flight track coloured by CO, along with its calculated

kriged plane and kriged variance plane relative to the original dataset is provided in Fig. 10.

$CH_4$, $CO_2$ and CO fluxes (Table 3) can be compared to a previous study by O'Shea et al using a similar mass balance

approach. CO fluxes are found to be ~half those calculated by O'Shea in the summer of 2012, whereas the $CH_4$ flux

calculated here is double. Our $CO_2$ flux estimate is within 10% of O'Shea's. When considering these data, one should be

mindful that aircraft measurements are representative of a single point in time and therefore cannot be aggregated over

longer periods. As such they are highly sensitive to meteorology and activity at the time of measurement, and the

methodology used for processing.

Of particular methodological importance is the criteria used to define the background. The impact this choice has in

determining which emissions contribute to the measured mass balance flux has been the subject of a recent study based on

the INFLUX project (Turnbull et al., 2018). In the case of flight C016, due to the development of the boundary layer during

the times between the upwind and downwind legs, upwind measurements were not considered representative of the

downwind background. Instead measurements from the downwind leg, outside of the plume, were used (as employed by

Turnbull et al., 2018). This is a different approach to the upwind background used by O'Shea et al (2014), therefore the

measured fluxes correspond to aggregate emissions from different areas. This could explain much of the discrepancy in

results between the two studies.

The difficulty in defining an emission aggregation area for mass balance flights around London, for any choice of

background criteria, has been discussed in depth by Pitt et al. (2019). In that study, mass balance fluxes from a different case

study flight around London (conducted in 2016) were found to be biased high compared to the results of a simple transport

model inversion using the same aircraft data, if the mass balance fluxes were assumed to represent only emissions from

Greater London. The mass balance flux estimates from that study are given in Table 3; these were also calculated using a

downwind background but due to differences in prevailing wind direction they capture emissions from a difference area with

respect to both this work and the results from O'Shea et al. (2014). The best way to design aircraft sampling strategies and

process the data to determine bulk emissions from megacities is the subject of ongoing discussion and research.

### 3.3.2 Pollution plumes from different local land sources: Flights C017 & C018, 4$^{th}$ July

For C017-18, there were also clear differences between the composition of the air sampled upwind (flight segments 3 and 2-

4 respectively) and downwind (segments 6-11 and 7-11 respectively) indicating different emission sources for the air masses

sampled either side of the city. During both flights, the pollution encountered downwind was more dispersed than the

previous day and exhibited a very different profile. However, there were also distinct differences between the composition of



the upwind samples between the morning and afternoon flights suggesting different air mass origins.

Upwind measurements from flight C017 showed very low levels of CO, $O_3$ and particles (mostly <88 ppbv, <35 ppbv, <2500 $cm^{-3}$ with periodic spikes) compared with flight C016, indicating much cleaner background air. $NO_x$ levels were slightly higher though (mostly ~1.0 ppbv with multiple peaks above 2.5 ppbv), suggesting a larger contribution from local emission sources than on the previous day. This fresh $NO_x$ likely also contributed to the reduced $O_3$ concentrations through NO titration. While the total concentrations of VOCs from the four WAS collected along this segment correlate well with

other pollutants ($r^2$ = 0.85, 0.99, 0.84 and 0.79 against CO, $NO_x$, $CH_4$ and aerosol number density respectively), acetylene which has an atmospheric lifetime of ~2-3 months against a typical OH concentration of ~$10^{-6}$ molecules $cm^{-3}$ is not well correlated with $NO_x$ ($r^2$ = 0.48) although it is against the longer-lived pollutants ($r^2$ = 0.92, 0.88 and 0.74 against CO, $CH_4$ and aerosol number density). This is typical of transported air (McMeeking et al., 2013), further confirmation that we were sampling aged background air mixed with some local fresh emissions. Back-trajectories (top panels of Fig. 11) show winds

were blowing from the west and north during this flight bringing relatively clean air to the region. This is further corroborated by a high altitude leg during the reciprocal runs over East Anglia (flight segment 12 on Figs. 1 and 6) downwind of London. Along this leg, which at a height of just under 2km was well above the BL, concentrations of gas-phase pollutants were all lower than those sampled in the upwind BL (~10s pptv of $NO_x$, CO ~80 ppbv, $O_3$ ~26 ppbv) indicating the long-range transport of clean air into the region.

Total concentrations of VOCs were higher downwind (peaking at 4.1 ppbv) with the strongest enhancements in propane (max 0.89 *vs.* 0.25 ppbv) and n-butane (max 0.64 *vs.* 0.16 ppbv). WAS have previously been successfully deployed on the ground and from aircraft to complement real-time measurements and to identify sources (e.g. Tiwari et al., 2010; Breton et al., 2017; Aruffo et al., 2014; Warneke et al., 2013; Cain et al., 2017; Lee et al., 2018). Tiwari et al., (2010) reported high concentrations of ethane, propane, and n-butane in Yokohama, Japan, which they attributed to fugitive emissions from

petroleum refining and evaporation. Ethane, propane, n-butane and cyclopentane, exhibit the highest average concentrations across all three flights and can likely be similarly attributed to petrochemical refining and natural gas processing.

In contrast to C016, total VOC concentrations in the plume were most strongly correlated with $NO_x$ ($r^2≈0.97$). $O_3:NO_x$ is much lower (27.3 compared with 40.7 upwind), suggesting that downwind of London we were mostly sampling fresh local emissions. Benzene was well correlated with $CH_4$ ($r^2≈0.96$) and aerosol number density ($r^2≈0.92$) but less with $NO_x$

($r^2≈0.71$), whereas toluene showed only weak correlation with all continuous measurements. One possible interpretation is that local sources of benzene include a mix of vehicle and industrial (e.g. natural gas processing and petrochemical refining) emissions, while additional toluene emissions originate from non-fossil fuel related industries, in particular solvent processing and use and brewing (e.g. NAEI, 2015; Gibson et al., 1995). Toluene emissions have a strong solvent component with no corresponding benzene emission. Data from the NAEI for VOCs indicate there has been a relative increase over the

last decade in the contribution solvents to toluene emissions, changing the source profile for benzene and toluene. This, taken in conjunction with our findings that local sources can strongly mediate benzene:toluene ratios on small spatial and temporal scales, suggest that their use in identifying the age of urban plumes may be more limited than previously assumed.

Further evidence that the pollution sampled in the plume is predominantly derived from local sources comes from the profile of VOCs by altitude. Unlike the London outflow plume sampled on 3[rd], the highest concentrations were recorded during the

run at the lowest altitude (Fig. 12a).

In contrast to the morning flight, the back-trajectories for the afternoon flight, C018 (bottom panels of Fig. 11), show a mix of airmass origins. While a large proportion of the air also arrives from the west and north, there is a substantive contribution from the west-south-west, along a similar trajectory to that for flight C016. This rather neatly explains our upwind atmospheric measurements lying between those of the two other flights, C017 with clean air from north and west, and C016





with high aerosol number density and CO from strong pollution sources to the southwest. $NO_x$ concentrations are elevated along this segment with local sources strongly contributing to the pollution sampled here as would be expected given the slower wind speeds on 4th July.

WAS collected during C018 show many similarities with those collected during the morning (C017). Again, total VOC concentrations are higher downwind than upwind of London (3.0 *vs.* 1.8 ppbv) with the highest relative increases in propane
and n-butane (8 to 14% and 6 to 8% respectively). The changes are smaller though, suggesting that although we were sampling emissions from the same local industries during both flights, the fresh emissions were mixed with more aged background air in the afternoon. Absolute and proportional concentrations of isoprene, which is mainly emitted from biogenic sources, were far higher during the afternoon than the morning, as expected given the strong dependence of isoprene emission rates on light and temperature (e.g. Guenther et al., 1991; 1995). Although $O_3$:$NO_x$ ratios were reduced to
21.6 in the plume (from 45.2 upwind), the highest concentrations of $O_3$ (up to 48 ppbv) of any of the flights were measured during the downwind legs of this flight in spite of the relatively high $NO_x$ (average mixing ratio of 2.0 ppbv, peaking at ~5 ppbv). VOC concentrations in the plume were strongly correlated with CO and $CH_4$ ($r^2 \approx 0.96$ and 0.93 respectively) but showed no correlation against either $NO_x$ ($r^2 \approx 0.07$) or aerosol number density ($r^2 \approx 0.44$). The high $NO_x$ levels observed in the plume suggest that local sources are contributing strongly while the high $O_3$ and correlation of VOCs with long-lived
pollutants is indicative of more aged (polluted) air from the south-west.

The lowest WAS sampling altitude during C018 was ~400 m which makes a direct assessment of the relative contributions of local to transported pollution difficult. In contrast to the morning flight, higher concentrations of VOCs appear to occur at higher altitudes (see Fig. 12b) resulting from a combination of stronger vertical mixing during the afternoon and the influence of long-range transport. Unlike the previous day, however, concentrations increased with altitude to the top of the
BL (at >1 km) suggesting we were sampling well-mixed pollution originating from both local (low-level) and distant sources, rather than a still-distinct relatively local London plume as in C016.

A particularly interesting feature of the reciprocal runs for both flights C017 and C018 was the presence of two spatially and chemically distinct spikes of pollution, which we refer to as the "West plume" and "East plume". The West plume was observed in the same location during both morning and afternoon; the East plume was slightly further (~11km) to the south
and east in the morning, consistent with the back-trajectories (Fig. 11) which show recirculation from the North Sea coast and suggest that aside from the afternoon East plume influence from London outflow was minimal.

Table 4 provides a summary of the average and peak concentrations for the full leg and the West and East plume for each reciprocal run and shows that, although not separated far in space or time, the two plumes were chemically distinct at all heights and for both flights. The composition of each plume was consistent across time, although concentrations were
generally lower in the morning. Concentrations of CO and $NO_x$ and aerosol number density were elevated in the West plume relative to the background by as much as 3 ppbv (~3%), 1.7 ppbv (>100%) and $10^3$ cm$^{-3}$ (15%) in the morning and ~15%, ~100% and 20% in the afternoon. By contrast, only $NO_x$ was found to be consistently high in the East (by as much as 200% during both flights), suggesting different source sectors or differences in air mass origins between the plumes.

Concentrations of $CH_4$ varied little either spatially or temporally across the reciprocal runs or plumes. Although slightly
enhanced near the surface, differences were <1% suggesting that local sources contribute little to atmospheric $CH_4$ concentrations in the region.

During both morning and afternoon reciprocal runs, average CO concentrations were higher in the West plume than the full flight leg but lower in the East (by up to 3 ppbv or ~3%), indicating a strong source toward the western end of the flight track. Vertical distributions were similar across the full leg and both plumes in the morning with highest levels observed at
674m (peaking at 108.2 ppbv in the West plume), suggesting concentrations were dominated by transported air from more industrial areas to the west and north (Fig. 11). During flight C018 however, peak CO occurred at lower altitudes in each of





the plumes (127.7 ppbv at 686m in the West and 104.5 ppbv at 553m in the East) than the leg as a whole (843m). We observed CO enhancements as high as 23% (in the afternoon in the West plume) with the maximum enhancement at an altitude coinciding with the maximum absolute concentrations. The exception to this is the East plume where the maximum

peak enhancement occurs at the lowest altitude during C017; strong CO enhancements were also observed at this level in the East during the afternoon. These observations are consistent with our trajectory analysis (see Fig. 11) that the eastern end of the reciprocal runs receives a flow of (relatively) clean air from the north resulting in a lower background than the western end with long-range transport bringing more polluted air from the west. However, the pronounced peaks in CO at relatively low altitudes suggest that there are also substantial local sources.

$NO_x$ is relatively short-lived so observed concentrations, which were highly variable in space and time, reflect localised sources rather than long-range transport. During flight C017, the highest $NO_x$ levels were observed along the lowest run (4.85 ppbv at 263m) in the East plume but further aloft at 674m (3.73 ppbv) in the West plume. The maximum increases in $NO_x$ also occurred at 263m in the East (>200%) and at 830m in the West (~140%). Mixing ratios were generally ~40% higher in the afternoon than morning, but spatially, the pattern was repeated with peak enhancements at the surface in the

East (~190%) and aloft (998m) in the West (~150%). Interestingly, while there was a rapid decline in concentration with altitude in the East plume during both flights, concentrations were relatively constant throughout most of the mixed layer in the West plume. This, together with the higher ratios of $NO_x$ to CO in the East plume, is indicative of strong sources in the immediate vicinity while more distant sources mix with freshly emitted $NO_x$ to enhance the concentrations at higher altitudes in the West plume, where $NO_x$ to CO is relatively low.

Table 4 also shows evidence of $NO_x$ titration of $O_3$ in both plumes during the afternoon flight, most pronounced in the East plume and at the lowest altitudes where $NO_x$ levels were highest. Outside of the plumes, $O_3$ mixing ratios were relatively constant at ~40 ppbv in the morning and slightly higher (~44 ppbv) in the afternoon, as expected for a secondary pollutant formed as a product of the photochemistry. Near the surface in the C018 East plume, $O_3$ dropped by ~3 ppbv (~8%) due to direct reaction with NO (NOx levels reached >10 ppbv).

Aerosol number density (column "CPC" in Table 4) was consistently highest at the surface, falling from >7400 at 263 m to 5700 $cm^{-3}$ at 831 across the full flight leg in the morning and >5700 to 5300 $cm^{-3}$ at 1155 m in the afternoon. This is consistent with fresh emissions of small particles coalescing and coagulating to form a smaller number of larger particles as they are mixed and lofted. Morning number concentrations were substantially higher than in the afternoon (>25% higher in the West plume and ~10% in the East). This is likely due to the trapping of particles in the stable nocturnal BL and the

dilution effect of the increasing mixed layer depth over the course of the day. Number density was much higher in the West than the East plume during both flights. The largest increase in number in the West plume occurred near the surface (altitudes up to 522m) in the morning and at 283m in the afternoon. AMS data, only available for the afternoon flight C018, further supports this apparent difference in emission source and strength between the eastern and western ends of the reciprocal runs with the West plume showing an increase in PM1 due to high levels of organic and nitrate aerosols.

By combining our back-trajectories for airmasses sampled in each plume with UK NAEI data for the region, we were able to identify local point sources to which the observed West plume is likely to be attributable. For CO, $NO_x$ and PM1 we calculated a "source intensity" at the point of interception based on an assumption that concentrations decayed with distance from source by an inverse square law (i.e. assuming a zero wind dispersion and neglecting chemical transformation).

The largest local contributions to CO in the West plume were power stations at Thetford and Ely in the morning, but the

slight change in wind direction in the afternoon resulted in large additional contributions from local construction and food and drink manufacturers. Interestingly, it was the same point sources that made the biggest contribution to $NO_x$ in the West plume. The only likely major local source of CO in the East plume was British Sugar and that was only directly upwind during the afternoon. There was no obvious point source affecting $NO_x$ concentrations in the eastern end of the reciprocal





runs and we speculate the very high levels observed in the East plume are the result of traffic emissions, particularly from the junctions between the major A144, A146 and A143 roads which were almost directly overflown.

Landfill gas combustion and brick manufacturing were likely the principle local sources of PM1 throughout the day across both plumes. Power stations again contributed strongly to the West plume and probably account for the high nitrate component of the fine particles observed in this plume, while landfill gas combustion and emissions from British Sugar are high in organic matter. There were fewer (and weaker) sources at the eastern end of the reciprocal run resulting in the low
aerosol number density observed.

### 3.4 Marine emission sources

In addition to the clear pollution events described in the previous section, we observed substantial spikes in concentrations during low-level flight legs over the sea. While it was difficult to positively identify the sources of the pollution observed over the land surface due to the complex interactions of photochemical processing and atmospheric dynamics, we were able
to directly attribute some of the peaks observed in the marine BL to specific vessels. We describe one such situation here.

Between around 10:21 and 10:27 UTC on 4th July (flight C017) FAAM BAe-146 overflew the Dover Straits at an altitude of between 24-75 m. Clear spikes in pollutant concentrations and aerosol number density were directly seen on most on-board instruments and we were able to observe the passage of a number of large ships which appeared to correlate with these enhancements. In order to evaluate whether a part of the observable signal in the different variables was attributable to
marine traffic, we plotted the time series of $NO_x$, NO, $NO_2$, and $CO_2$ concentrations and the aerosol number density (CPC; Fig. 13). We identified a number of plumes throughout this portion of the flight but focused our analysis on the clear sharply defined peak in concentration observed at 10:22:30 and marked with an 'X' on Fig. 14. At this point, $NO_x$ levels were observed to be elevated by a factor of ~20 and aerosol number density by a factor of ~5.

As marine emissions are known to be an important source of both $NO_x$ and PM (Corbett et al., 1999), we used data obtained
from Marine Traffic (https://www.marinetraffic.com) to examine the vessels navigating this area at the time of overflying. Fig. 14 maps the paths of those ships with a tonnage >10 kton (thin coloured lines, with colour ranging from purple to yellow corresponding to specific times between 10:13:20 and 10:26:40), overlaid with the path of the aircraft (thick line); arrows denote windspeed and direction. Only the portion of the flight path above the sea is shown in Fig. 14 with further detail of the intersection of the aircraft with the main plume shown in Fig 15.

The 'X' in Fig. 14 and the 'X' labelled with 'A' in Fig. 15 correspond to the location of the prominent peaks seen in Fig. 13b-d. At this point, a large ship had passed under the flight path shortly ahead of our transit. The second line in Fig. 15 parallel to the ship trajectory denotes the location of the plume emitted 40 s downwind of its position. We were able to identify this vessel from Marine Traffic data as a 15 kton Liberian container ship (see Fig. 15). Other smaller plumes seen in Fig. 13 could not be directly attributed to a single ship and are likely an accumulation of emissions from a number of smaller
or more distant vessels (observable in Fig. 14).

### 4. Conclusions

We report here measurements of atmospheric conditions and composition made during 3 research flights from the UK's FAAM BAe-146-301 Atmospheric Research Aircraft over the course of two days in July 2017. Conditions were favourable for all flights and a change in windspeed and direction overnight enabled us to sample contrasting pollution events.

On 3rd July, moderate west-southwesterly winds produced a narrow distinct plume of pollution outflowing London. The clear edges and strong enhancement of the plume allowed us to apply a mass balance approach to estimate emissions of long-lived pollutants from the urban area. Our calculated fluxes of $CO_2$ agreed well with those previously reported for 2012



by O'Shea et al. (2014) but our estimated emissions of CO were a factor of 2 lower and CH$_4$ a factor of 2 higher. These
differences between campaigns are likely due to differences in methodology and the inherent sensitivity of the mass balance

method to the surface that has been sampled. Methods that can provide improved quantification on surface interaction are of
greater use when the emission source is not distinct from its surroundings.

The second and third flights on 4$^{th}$ July experienced much lighter and more variable winds with the result that pollution was
more widely dispersed and derived from a mixture of sources. In general, there was a strong contribution of fresh emissions
from local point sources with evidence of mixing with air transported from further afield bringing more aged pollution to the

region. We observed clear pollution events over northern East Anglia during both flights and flew a series of reciprocal runs
to sample these peaks over the full altitude of the boundary layer. Continuous real-time measurements of long-lived gas-
phase and aerosol pollutants were supplemented with analysis of a range of organic compounds from whole air samples
taken during the reciprocal runs.

Based on different relative abundances of organic compounds and the ratio of O$_3$:NO$_x$ we were able to determine source

sectors and individual sources for the morning and afternoon pollution spikes. During the morning most of the transported
air mass was from the north and west, and therefore relatively clean, and the pollution was predominantly fresh emissions
from local food and drink and construction industries. By contrast, the air mass in the afternoon contained more aged
pollution from the south-west, although still very little from the London area. We were able to attribute local emissions to
the same sources as well as a contribution from power plants in the area. The high NO$_x$ concentrations observed toward the

eastern end of the reciprocal runs appeared to emanate from traffic at a series of major road junctions.

Importantly though, our observations of local pollution episodes on 4$^{th}$ July strongly suggest that the use of the ratio of
benzene to toluene concentrations to assess air mass age and emission source is unreliable when applied over small spatial
and temporal scales. The increasing numbers of sources that emit toluene alone result in heterogeneous ratios of benzene to
toluene emissions from different source sectors, whereas the use of concentration ratios is based on known constant relative

source intensities.

These three flights give a clear demonstration of the power of airborne remote sensing to inform and constrain bottom-up
source attribution and emissions inventories. They also provide further evidence that the factors that control the air pollution
buildup in the London area are various and multiple: local emissions, transport from distant sources, terrestrial and marine
emissions. It is necessary to consider and constrain all of these factors to understand the problem and to develop effective

mitigation and control strategies.

**Acknowledgements**

This work was funded by EUFAR (European Facility for Aircraft Research ) with the financial contribution from the
European Commission for the management of EUFAR2, under the grant agreement no. 312609. EUFAR AISBL provided
partial financial support for the publication costs of this paper. Funding for WAS collection and analysis was provided by

National Centre for Atmospheric Science [NCAS].

Airborne data was obtained using the BAe-146-301 Atmospheric Research Aircraft flown by Airtask Ltd, maintained by
Avalon Aero and managed by the Facility for Airborne Atmospheric Measurements [FAAM] Airborne Laboratory, jointly
operated by UKRI and the University of Leeds. We thank in particular Axel Wellpott for his assistance with collating flight
data, our incredible pilots and all of the FAAM staff and ground crew whose skills and support made these flights possible.

We are grateful for the input and advice we received from Alex T. Archibald and Michelle Cain. We acknowledge the help
we received with data collection and initial processing from other participants of the STANCO training school: M. Ardelean,





A. Bordas, G. Cann, L. Cappellin, P. Dominutti, M. Koehler, O. Lauer, G. Methymaki, F. Pardini, L.-A. Picui, A. Holubova Smejkalova, and from undergraduate summer research intern Joe Westwood.

Kirsti Ashworth is a Royal Society Dorothy Hodgkin Research Fellow and acknowledges support and funding from the
Royal Society (award no. DH150070).

**Data availability**

In keeping with UK funding agency requirements for access and transparency in research, all data described in this analysis are available at CEDA (Centre for Environmental Data Analysis: http://data.ceda.ac.uk/badc/faam/data/2017.

**Author contributions**

K. Ashworth, S. Bucci, P. Gallimore, J. Lee, B. Nelson, A. Sanchez Marroquín, M. Schimpf, and P. Smith participated in STANCO, designed flight paths, collected, processed, analysed and plotted data from flights. P. Di Carlo, R. Krejci, J. McQuaid organised and participated in STANCO and assisted with data collection and interpretation. James Lee participated in STANCO and assisted with data collection, processing and analysis. W. Drysdale, J. Pitt and J. Hopkins processed, analysed and interpreted data. All authors contributed to discussin, writing and editing the manuscript.

**Conflicts of interest**

The authors declare that they have no conflict of interest.

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





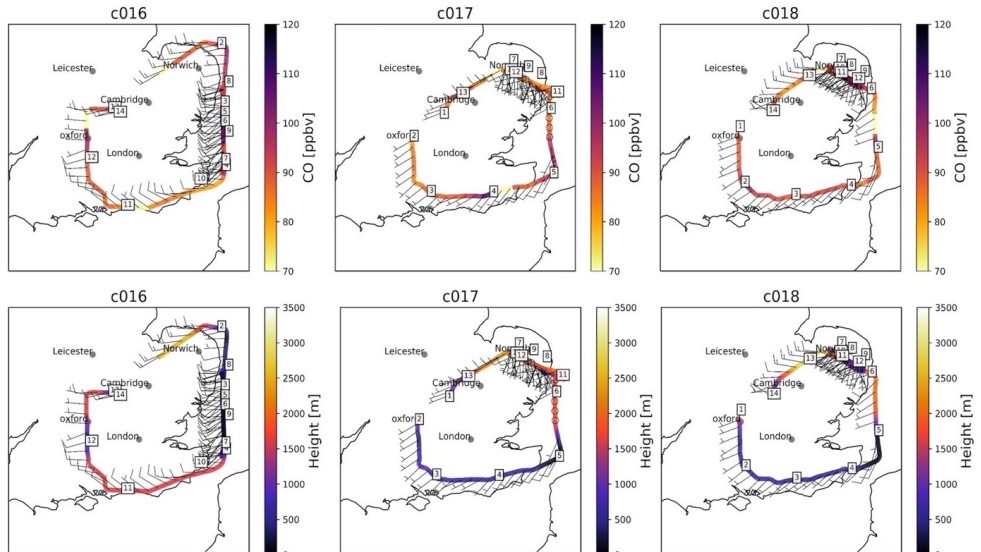

**Figure 1: Map of flight paths for all three flights (C016 on Monday 3rd and C017 and C018 on Tuesday 4th July 2017). Top panels show the concentrations of CO measured on board and the bottom panels FAAM BAe-146 altitude. Arrows indicate windspeed and direction at 1-minute intervals along the path. The numbers in boxes correspond to distinct flight segments which are used hereafter to locate FAAM BAe-146 geographically during the flight.**

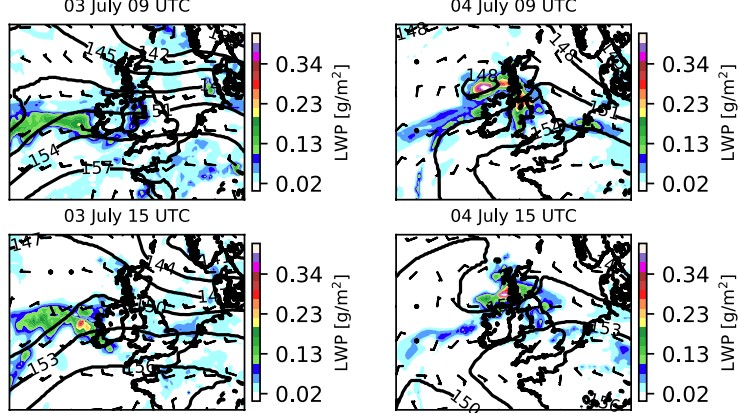

**Figure 2: Maps of geopotential height [dam], windspeed [kt], and liquid water path (LWP) [g/m²]; ERA-interim data.**

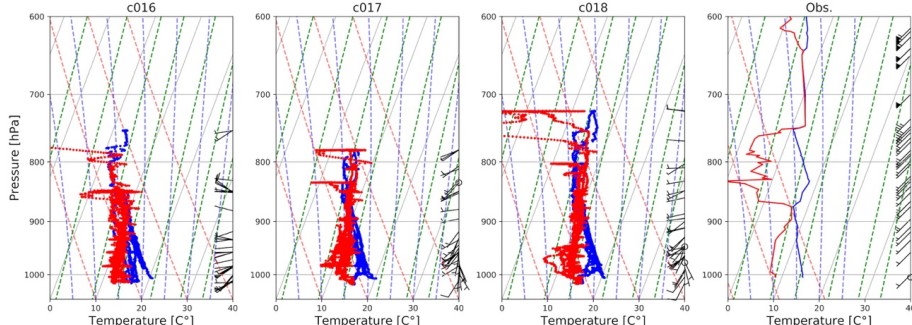

**Figure 3: SkewT-logP plots showing (a) temperature (blue) and dewpoint temperature (red) for all 3 flights, and (b) temperature and potential temperature from a radiosonde launched from Nottingham at 00:00 UTC on 4th July 2017.**





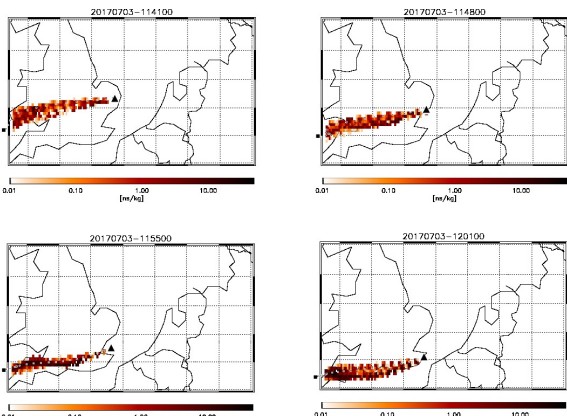

**Figure 4: FLEXPART modelled footprint of air mass arriving at the location of FAAM BAe-146 (black triangles) at four different positions along the reciprocal runs of flight C016. Each coloured pixel indicates the relative contribution of an inert tracer in that air to the total concentration of that tracer sampled on-board. The large black square shows the point of release of the air 24 hours prior to being intercepted by the aircraft. The dotted line of black and white squares shows the hourly weighted average trajectory of the air mass based on the relative contributions shown.**

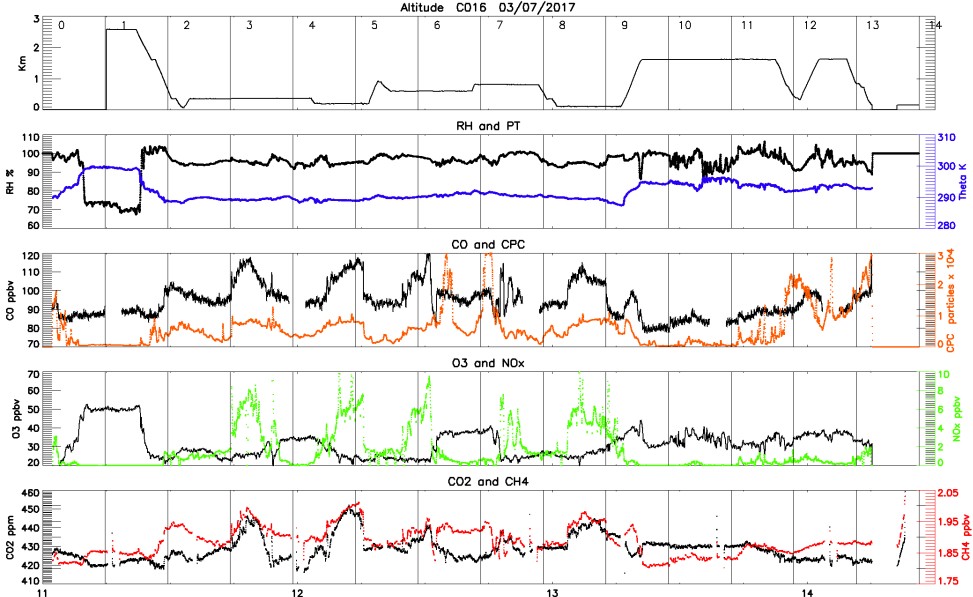

**Figure 5: Time series of the main observations during the first flight (3rd July, late morning). From the upper panel it is shown: the altitude of the flight, relative humidity (second panel, black) and potential temperature (blue), CO concentration (third panel, black) and CPC (aerosol number density, cm-3; orange), O3 (fourth panel, black) and NOx (green) concentrations, CO2 (fifth panel, black) and CH4 (red) concentrations. The numbered vertical lines correspond to the numbers along the flight path shown in Fig. 1.**





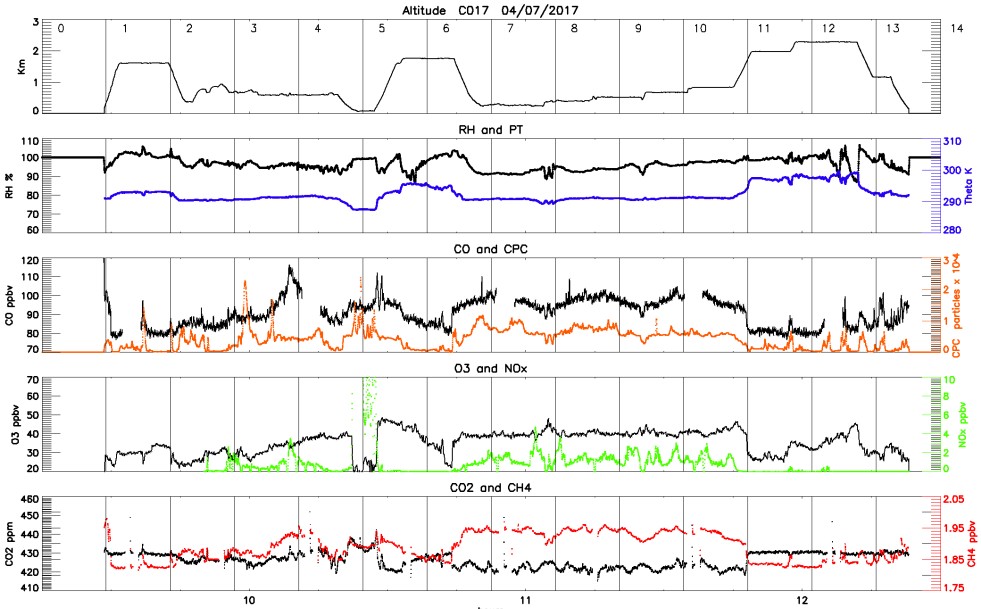

**Figure 6: As in Fig. 5 but for the second flight (C017; 4th July, morning).**

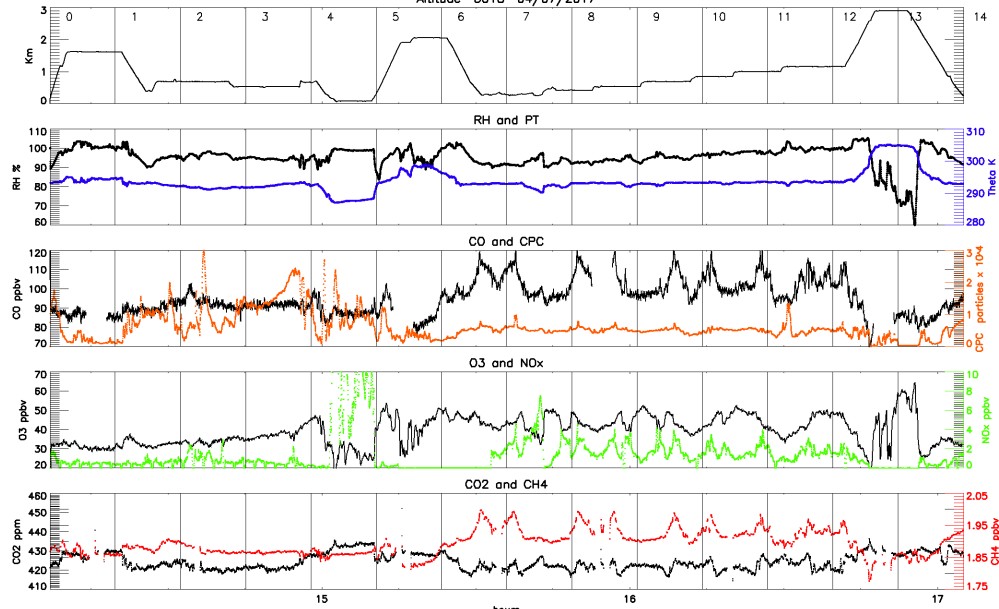

**Figure 7: As in Fig. 5 but for the third flight (C018; 4th July, afternoon).**





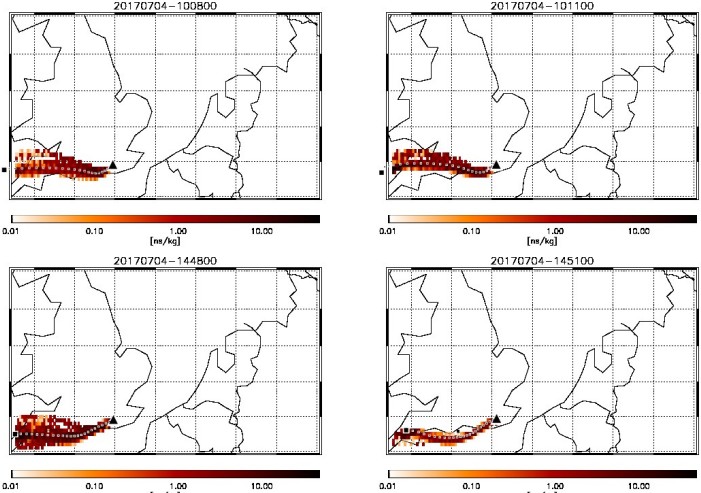

**Figure 8: FLEXPART modelled footprint of air mass arriving at the location of FAAM BAe-146 (black triangles) at 10:08 and 10:11 during flight C017 (top row) and at 14:48 and 14:51 during flight C018. Each coloured pixel indicates the relative contribution of an inert tracer in that air to the total concentration of that tracer sampled on-board. The large black square shows the point of release of the air 24 hours prior to being intercepted by the aircraft. The dotted line of black and white squares shows the hourly weighted average trajectory of the air mass based on the relative contributions shown.**

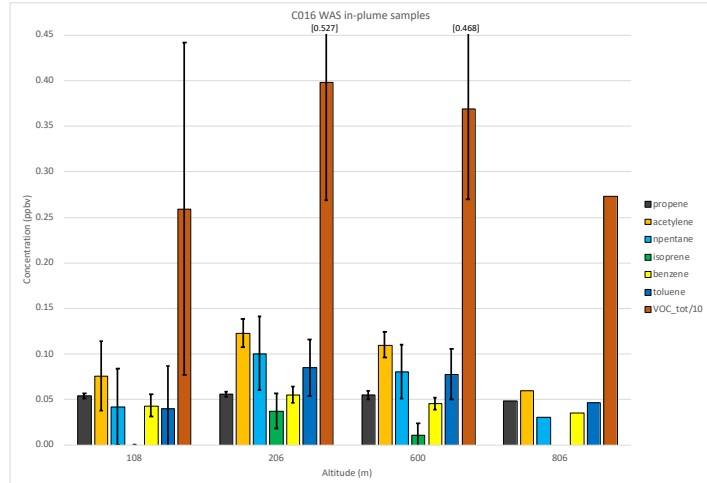

**Figure 9: Average concentrations of key VOCs (ppbv) collected via WAS during individual flight legs within the plume detected during flight C016. The average altitude of each flight leg is shown on the x-axis. Error bars denote ±1s.d.; numbers in square parentheses show top of error bars.**

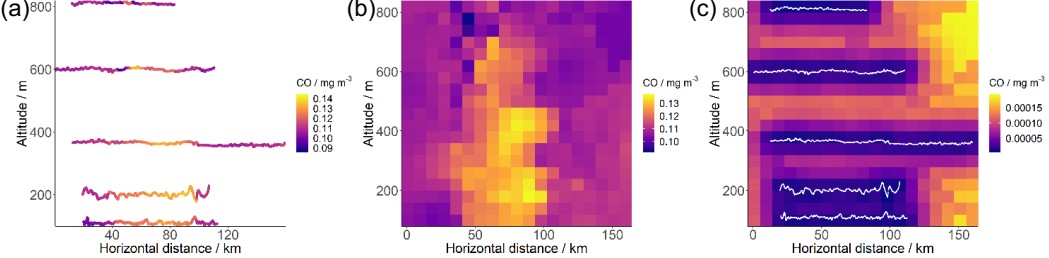

**Figure 10: Plots coloured by CO mass per volume (a) vertical plane of downwind measurements; (b) downwind flight tracks interpolated using kriging to produce a kriged plane; (c) variance map representing the kriging uncertainty, where the flight track is shown in white.**





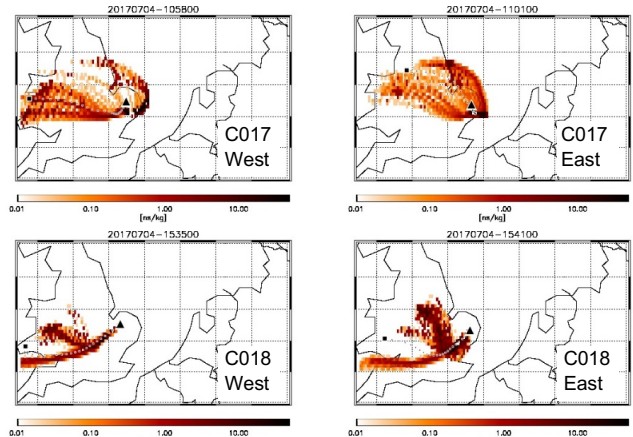

Figure 11: FLEXPART back-trajectories for air masses arriving at the location of FAAM BAe-146 as it intercepted the West (right column) and East (left) plume during the lowest of the reciprocal runs for flights C017 (top row) and C018 (bottom). Each coloured pixel indicates the relative contribution of an inert tracer in that air to the total tracer concentration sampled on-board. The large black square shows the point of release of the air 24 hours prior to interception. The dotted line of black and white squares shows the hourly weighted average trajectory of the air mass based on the relative contributions shown.

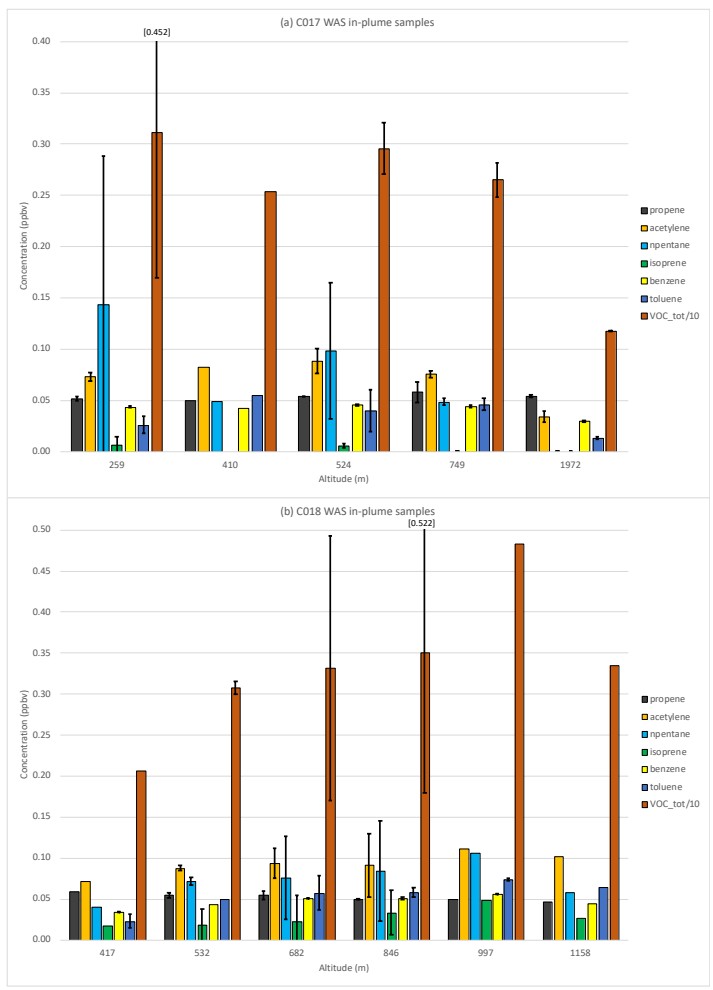





**Figure 12: Average concentrations of key VOCs (ppbv) collected via WAS during individual flight legs within the pollution plume detected during (a) flight C017, and (b) flight C018. The average altitude of each flight leg is shown on the x-axis. Error bars denote ±1s.d.; numbers in square parentheses show top of error bars.**

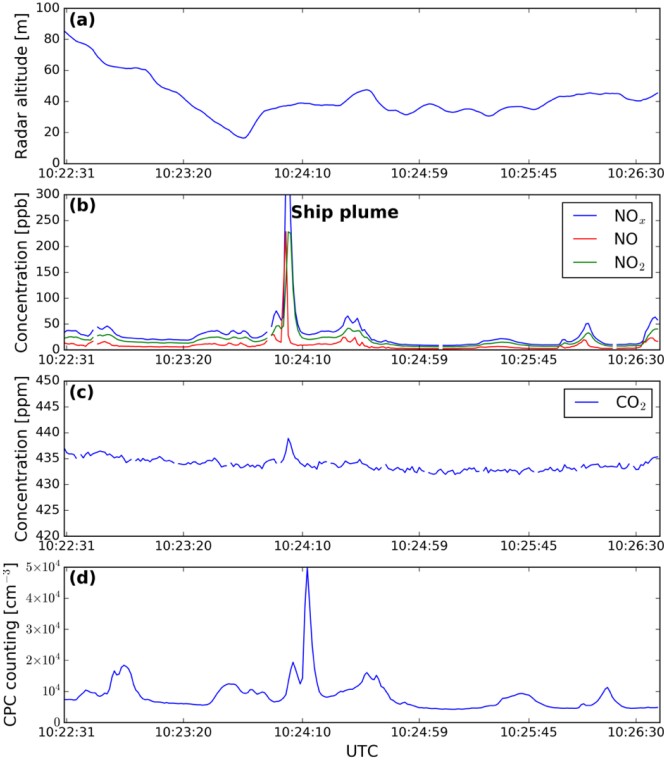

**Figure 13: Time series of different variables measured above the sea during the C017 flight: (a) altitude, (b) NO, NO$_2$ and NO$_x$ concentrations, (c) CO$_2$ concentration, and (d) total aerosol concentration. Only the times close to the large plume that could be correlated to a specific vessel are shown here. CO and O$_3$ concentrations (not shown) exhibited no enhancement. The plotted time period corresponds to the beginning of a level run at ~40m altitude, the trajectory of which can be seen in the Figs. 14 and 15.**

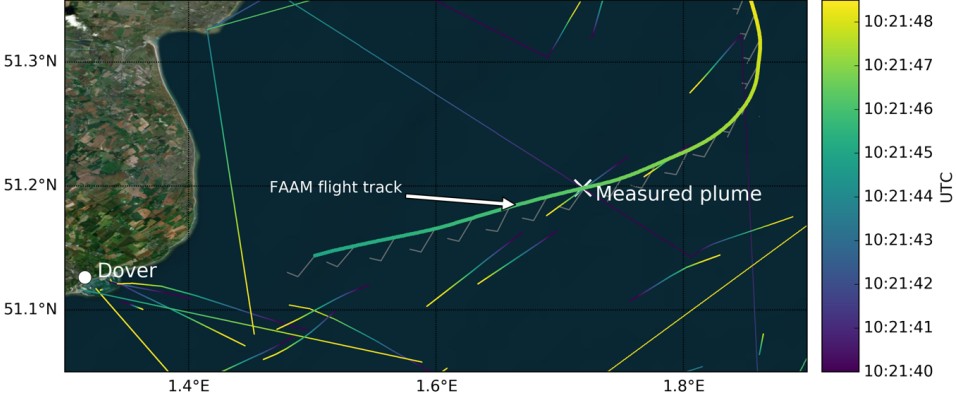

**Figure 14: Flight track of FAAM BAe-146 (thick colour line) during the C017 flight at the beginning of the straight level run at ~40 m a.m.s.l where the large ship plume was measured (indicated with a white cross). The end of the descent to 40 m was also included in the flight track. Wind speed and direction measured by FAAM BAe-146 are shown by the wind barbs. The trajectories of all the ships >10 kton are shown by thin colour lines with the colour scale denoting time. The base map was created using the World Imagery maps in ArcGIS® software by Esri. ArcGIS® and ArcMap™ are the intellectual property of Esri and are used herein under license. Copyright © Esri. All rights reserved. For more information about Esri® software, please visit www.esri.com.**



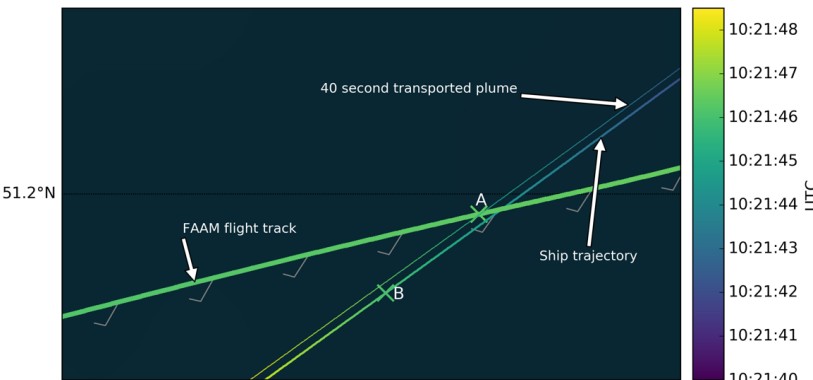

**Figure 15: Close-up of the flight track of FAAM BAe-146 and the ship to which the measured plume has been attributed. The second thinner line parallel to the ship trajectory is the position and time of the pollution plume emitted by the ship after being transported following the wind speed and direction for 40 s, the time delay between aircraft and ship trajectories. The plume produced by the ship at point B was detected by FAAM BAe-146 at point A 40 s later. The base map was created using the World Imagery maps in ArcGIS® software by Esri. ArcGIS® and ArcMap™ are the intellectual property of Esri and are used herein under license. Copyright © Esri. All rights reserved. For more information about Esri® software, please visit www.esri.com.**

| Flight ID | Date | Start time (UTC) | End time (UTC) |
|-----------|------|------------------|----------------|
| C016 | 3rd July | 11:10:24 | 14:03:07 |
| C017 | 4th July | 09:42:51 | 12:09:15 |
| C018 | 4th July | 14:20:59 | 16:53:29 |

**Table 1: Overview of flights.**

| | C016 | | | C017 | | | C018 | | |
|---|---|---|---|---|---|---|---|---|---|
| Species | Flight | Upwind | Plume | Flight | Upwind | Plume | Flight | Upwind | Plume |
| CO | 101.0 (9.4) | 94.79 (3.7) | 104.6 (9.3) | 92.26 (8.2) | 88.08 (8.4) | 99.39 (3.1) | 97.16 (9.2) | 92.18 (2.2) | 102.4 (6.7) |
| CH$_4$ | 2035 (26) | 2011 (2) | 2050 (28) | 2063 (24) | 2046 (14) | 2084 (5) | 2064 (23) | 2043 (2) | 2082 (20) |
| NO$_x$ | 2.13 (2.43) | 0.74 (0.20) | 3.12 (2.52) | 1.67 (4.26) | 0.81 (0.68) | 1.44 (0.61) | 1.95 (3.13) | 0.78 (0.48) | 1.96 (0.78) |
| O$_3$ | 28.87 (5.21) | 36.52 (1.14) | 26.16 (4.03) | 34.17 (6.54) | 32.64 (4.31) | 39.41 (0.44) | 36.65 (7.51) | 35.46 (0.67) | 42.48 (2.88) |
| CPC | 9930 (5940) | 19880 (2610) | 7210 (1890) | 5420 (3720) | 2590 (2160) | 7080 (810) | 7910 (5610) | 14870 (3930) | 5250 (510) |
| Total VOC | 3.13 (1.57) | 1.59 (0.27) | 3.74 (1.10) | 2.25 (0.83) | 1.76 (0.45) | 2.75 (0.25) | 2.49 (1.17) | 1.82 (0.08) | 2.97 (1.25) |
| ethane | 1.10 (0.33) | 0.78 (0.05) | 1.22 (0.25) | 0.91 (0.20) | 0.79 (0.10) | 1.04 (0.05) | 0.95 (0.32) | 0.77 (0.00) | 1.11 (0.32) |
| ethene | 0.14 (0.08) | 0.04 (0.01) | 0.17 (0.06) | 0.08 (0.03) | 0.08 (0.03) | 0.09 (0.01) | 0.09 (0.04) | 0.07 (0.01) | 0.09 (0.05) |
| propane | 0.38 (0.28) | 0.11 (0.03) | 0.51 (0.22) | 0.28 (0.23) | 0.15 (0.09) | 0.38 (0.11) | 0.33 (0.26) | 0.15 (0.02) | 0.42 (0.22) |
| propene | 0.06 (0.00) | 0.05 (0.00) | 0.06 (0.00) | 0.05 (0.01) | 0.052 (0.01) | 0.06 (0.01) | 0.05 (0.01) | 0.05 (0.00) | 0.06 (0.01) |
| iso-butane | 0.14 (0.12) | 0.03 (0.01) | 0.19 (0.09) | 0.07 (0.05) | 0.04 (0.03) | 0.10 (0.02) | 0.09 (0.07) | 0.05 (0.00) | 0.11 (0.07) |
| n-butane | 0.31 (0.26) | 0.07 (0.03) | 0.42 (0.20) | 0.18 (0.17) | 0.09 (0.08) | 0.25 (0.08) | 0.21 (0.17) | 0.10 (0.01) | 0.26 (0.17) |
| acetylene | 0.09 (0.03) | 0.05 (0.00) | 0.11 (0.02) | 0.06 (0.02) | 0.05 (0.01) | 0.08 (0.01) | 0.07 (0.03) | 0.06 (0.00) | 0.09 (0.02) |
| cyclopentane | 0.58 0.30 | 0.35 (0.11) | 0.62 (0.24) | 0.40 (0.12) | 0.35 (0.11) | 0.49 (0.13) | 0.45 (0.20) | 0.40 (0.10) | 0.55 (0.24) |





| | | | | | | | | | |
|---|---|---|---|---|---|---|---|---|---|
| iso-pentane | 0.15 (0.13) | 0.04 (0.02) | 0.20 (0.09) | 0.08 (0.06) | 0.05 (0.04) | 0.11 (0.02) | 0.10 (0.06) | 0.07 (0.01) | 0.11 (0.07) |
| n-pentane | 0.07 (0.05) | 0.02 (0.01) | 0.09 (0.04) | 0.06 (0.07) | 0.02 (0.02) | 0.07 (0.04) | 0.05 (0.04) | 0.03 (0.01) | 0.07 (0.04) |
| isoprene | 0.02 (0.02) | 0.00 (0.00) | 0.02 (0.02) | 0.01 (0.01) | 0.01 (0.02) | 0.00 (0.00) | 0.02 (0.02) | 0.02 (0.02) | 0.02 (0.02) |
| benzene | 0.05 (0.01) | 0.03 (0.00) | 0.05 (0.01) | 0.05 (0.01) | 0.05 (0.02) | 0.04 (0.00) | 0.04 (0.01) | 0.03 (0.00) | 0.04 (0.01) |
| toluene | 0.06 (0.04) | 0.02 (0.01) | 0.08 (0.03) | 0.04 (0.02) | 0.04 (0.02) | 0.05 (0.01) | 0.04 (0.03) | 0.03 (0.01) | 0.05 (0.03) |

**Table 2: Average concentrations of trace gases (ppbv) and aerosol number density (CPC; cm$^{-3}$) for whole flight, upwind segment and downwind curtain runs at an altitude corresponding to the upwind leg for each flight. Numbers in parentheses show ±1 s.d.**

| Species | This work / mol s$^{-1}$ | O'Shea et al. / mol s$^{-1}$ | Pitt et al. / mol s$^{-1}$ |
|---|---|---|---|
| $CH_4$ | 431 ± 59 | 238 ± 12 | 182 ± 9 |
| $CO_2$ | 32,176 ± 8,890 | 35,861 ± 2,553 | 44,700 ± 1200 |
| CO | 116 ± 17 | 219 ± 8 | 178 ± 6 |

**Table 3: Initial fluxes from Greater London determined using the mass balance approach in this study and compared to those found by O'Shea et al. (2013).**


**C017**

| Run | Ave. alt. (m) | Full flight leg [CO] (ppbv) | [CH₄] (ppbv) | [NOₓ] (ppbv) | [O₃] (ppbv) | CPC (cm⁻³) | "West plume" (52.57N, 1.00E) [CO] (ppbv) | [CH₄] (ppmv) | [NOₓ] (ppbv) | [O₃] (ppbv) | CPC (cm⁻³) | "East plume" (52.42N, 1.45E) [CO] (ppbv) | [CH₄] (ppmv) | [NOₓ] (ppbv) | [O₃] (ppbv) | CPC (cm⁻³) |
|---|---|---|---|---|---|---|---|---|---|---|---|---|---|---|---|---|
| 4 | 263 | 95.59 | 2.085 | 1.509 | 39.39 | 7415 | 96.68 (101.6) | 2.082 (2.103) | 1.324 (2.080) | 39.08 (40.86) | 8517 (13800) | 94.17 (102.2) | 2.082 (2.094) | 2.411 (4.855) | 39.36 (41.85) | 5874 (6660) |
| 5 | 408 | 97.04 | 2.084 | 1.401 | 40.27 | 6969 | 99.48 (105.6) | 2.089 (2.100) | 1.520 (2.078) | 39.67 (40.94) | 8083 (9650) | 94.38 (99.48) | 2.079 (2.092) | 1.954 (4.476) | 40.69 (43.54) | 5754 (10300) |
| 6 | 522 | 97.29 | 2.083 | 1.519 | 40.09 | 6664 | 100.1 (107.2) | 2.090 (2.097) | 1.579 (2.233) | 39.82 (40.99) | 7851 (12700) | 94.90 (101.1) | 2.078 (2.091) | 2.013 (2.908) | 40.09 (43.45) | 5760 (8090) |
| 7 | 674 | 97.85 | 2.083 | 1.415 | 40.23 | 5936 | 100.5 (108.5) | 2.087 (2.097) | 1.704 (3.187) | 39.39 (41.46) | 6269 (8570) | 95.15 (101.3) | 2.076 (2.086) | 1.391 (3.182) | 40.84 (42.75) | 5763 (23100) |
| 8 | 831 | 97.13 | 2.079 | 1.277 | 41.04 | 5682 | 100.7 (106.3) | 2.083 (2.092) | 1.804 (3.063) | 39.88 (42.32) | 5581 (7000) | 95.14 (99.06) | 2.073 (2.085) | 1.291 (1.928) | 42.40 (44.01) | 5915 (7020) |

**C018**

| Run | Ave. alt. (m) | Full flight leg [CO] (ppbv) | [CH₄] (ppbv) | [NOₓ] (ppbv) | [O₃] (ppbv) | CPC (cm⁻³) | "West plume" (52.57N, 1.00E) [CO] (ppbv) | [CH₄] (ppbv) | [NOₓ] (ppbv) | [O₃] (ppbv) | CPC (cm⁻³) | "East plume" (52.46N, 1.30E) [CO] (ppbv) | [CH₄] (ppbv) | [NOₓ] (ppbv) | [O₃] (ppbv) | CPC (cm⁻³) |
|---|---|---|---|---|---|---|---|---|---|---|---|---|---|---|---|---|
| 5 | 287 | 105.1 | 2.085 | 2.785 | 41.52 | 5754 | 111.9 (128.1) | 2.102 (2.116) | 2.498 (5.686) | 43.27 (45.76) | 6777 (19300) | 97.92 (107.7) | 2.067 (2.080) | 4.157 (7.760) | 38.50 (47.32) | 5222 (6450) |
| 6 | 415 | 105.2 | 2.084 | 2.414 | 43.03 | 5460 | 112.5 (126.0) | 2.101 (2.116) | 2.624 (4.919) | 43.46 (47.88) | 5984 (9480) | 97.27 (104.3) | 2.067 (2.079) | 2.909 (4.177) | 41.09 (47.38) | 4949 (6150) |
| 7 | 533 | 101.8 | 2.077 | 2.150 | 44.09 | 5388 | 116.2 (123.8) | 2.097 (2.116) | 2.440 (4.174) | 42.47 (45.54) | 5741 (8420) | 99.09 (115.6) | 2.067 (2.082) | 2.697 (3.780) | 42.20 (49.48) | 5247 (6370) |
| 8 | 686 | 103.6 | 2.079 | 1.993 | 43.96 | 5167 | 110.5 (127.7) | 2.093 (2.114) | 2.404 (4.066) | 43.09 (47.82) | 5476 (8390) | 98.66 (104.5) | 2.068 (2.074) | 2.172 (4.919) | 43.28 (47.51) | 5159 (6010) |
| 9 | 843 | 103.8 | 2.081 | 1.835 | 44.11 | 5077 | 107.6 (122.0) | 2.087 (2.112) | 1.985 (3.550) | 41.61 (45.51) | 4821 (7370) | 99.12 (104.5) | 2.071 (2.080) | 1.830 (2.780) | 46.12 (49.47) | 5411 (7000) |
| 10 | 998 | 103.4 | 2.078 | 1.718 | 44.08 | 5103 | 108.7 (125.0) | 2.086 (2.112) | 2.311 (4.409) | 41.83 (46.14) | 4758 (6440) | 101.2 (107.8) | 2.076 (2.082) | 1.441 (1.774) | 48.74 (50.57) | 5591 (6430) |
| 11 | 1155 | 102.6 | 2.075 | 1.544 | 41.29 | 5305 | 103.4 (114.9) | 2.075 (2.100) | 1.733 (2.809) | 40.12 (42.48) | 3914 (5110) | 104.8 (113.9) | 2.082 (2.094) | 1.677 (2.619) | 47.96 (49.30) | 5193 (6440) |

Table 4: Overview of measured concentrations of CO, CH₄, NOₓ and O₃ in and aerosol number density (CPC) measured during the reciprocal runs on flights C017 (top) and C018 (bottom). The full flight leg values are averages along each run, the West plume are average and maximum (in parentheses) concentrations within the West plume and the same for the East plume. The average altitude is shown for each leg; the average latitude and longitude of the centre of each plume shown for West and East plumes.