# Peer review of "Megacity and local contributions to regional air pollution: An aircraft case study over London"

_Atmospheric Chemistry and Physics, 2019_

## Referee Comment (RC1) · Anonymous Referee #1 · 28 Nov 2019

Summary:

This paper discusses concentration measurements for several air pollutants (CO, VOCs, NMHCs and PM), and greenhouse gases ($CO_2$, $CH_4$), and derives Greater London emission fluxes for $CH_4$, $CO_2$, and CO, from aircraft sampling in a series of flights around South East England in July 2017. It uses world-class research facilities (the NERC FAAM aircraft) and calibrated instrumentation and combines the measured geospatial dataset with back-trajectory modelling and simple mass balancing approaches to discuss airmass history and emission sources. The focus is on quantitative emissions from London as a megacity, but the paper also discusses localised and spatially-contrasting sources qualitatively. The use of tracer:tracer ratios for pollution source attribution and ageing is very interesting and offers some very useful guidance

for those following this work. Put together, this represents an excellent and rigorously compiled dataset, analysed using appropriate modelling and analysis methods, resulting in some incremental conclusions and comparisons with similar previous studies. The paper is rigorous and of high quality in terms of its presentation and analysis. I believe it would be of interest to readers of ACP, especially for those with interests in air quality emissions and chemistry and GHG flux methods. There are just a few suggestions to further improve the manuscript that the authors could consider below, and a couple of potentially important clarifications that may or may not require slight modifications to the analysis.

Specific comments: 1/ The paper compares CH4, CO2 and CO fluxes derived using mass balancing with those reported using highly analogous aircraft sampling and methods by O'Shea et al., 2014 and Pitt et al., 2019. The paper currently discusses the relative quantitative differences with those studies. A further useful dimension to the discussion surrounding that comparison could be more thought as to "why" they differ. Line 355 suggests that the method for defining the background concentrations for mass balancing may be a potential reason for discrepancy between O'Shea and this study, and that the spatial footprint of emissions may be different. I would agree that that is one potential source of difference. However, an equally valid reason is that the emissions from London are simply different at these two very different times. Would it possible to raise some hypotheses on the reasons for truly different emissions? Would a comparison with the NAEI inventory help to elucidate the changing sources with time between 2014 and 2017? If the authors do not wish to go to this next level of interpretation, then simply stating that the emissions are likely to vary with time would go some way to ensuring the reader is left with a more correct (or full and balanced) conclusion. In other words, I don't believe that difference in method and footprint solely explain the different fluxes, which is the message that is perhaps currently conveyed.

2/ The paper often refer to "spikes" or "spiking". I think words such as "concentration enhancement" or "transient enhancement" could be more intuitive.

ACPD

3/ All concentrations are reported in units of ppv (e.g. ppbv) in the paper. As far as I understand, most in situ instruments on the FAAM aircraft report molar mass concentration (e.g. ppb). The GHG kit certainly does. This does make a difference for flux calculations as ppbv is not equal to ppb and a molar ratio must be used to convert between the two units prior to use in the mass balance equation. Has this been done? If not, the fluxes may need to be corrected. Otherwise, please confirm how ppb etc was converted to its volumetric equivalent for CO, $CO_2$, and $CH_4$.

Technical comments: 1/ Some units do not have a space between quantity and unit, e.g. 450m (and elsewhere where m are used). 2/ Figure 3 – does not appear well on my screen. Can the axes lines be thickened? 3/ Legend on Figure 12 – hard to read – can a larger font size be used?

---

## Referee Comment (RC2) · Anonymous Referee #2 · 10 Dec 2019

Review of "Megacity and local contributions to regional air pollution: An aircraft case study over London," Ashworth et al., ACP (2019)

Summary

Ashworth et al. Present observations of atmospheric composition from a series of three flights conducted around London in July 2017. They apply a combination of trajectories and correlation analysis to illustrate the emission and transport of pollution from both London and other point sources in the area. The topic is appropriate for ACPD. The English is good. There are a few more figures than are necessary to convey the information.

This paper is largely a description of the observations, with some analysis of emissions that carries large caveats and uncertainties. These particular measurements have not been presented before, but there have been previous similar observations. It is not clear to the reviewer what new or novel things we have learned, or can learn, from this dataset. Some reorganization would help, as would focusing more on current open questions.

Specific Comments

Use of WAS data for source identification: there are many examples in the literature of the use of specific hydrocarbon ratios to identify distinct emission sources. Why was this not done here?

Flight segments: It is helpful to number flight segments when comparing the map (Fig. 1) and time series (Figs 5 – 7). However, there are so many flight segments that the map looks cluttered and it is sometimes difficult to identify features discussed in the text. Rather than using one for every 5 minutes, how about 4 – 5 sections per flight?

L342 onward: discussion seems to indicate that the flux estimates shown in Table 3 are not comparable because of differences in the methodology and data used to do the calculation. Is this really the case, or is it just that the footprint is different? Also this makes Table 3 itself questionable, as it is comparing numbers that are not comparable. Statements regarding the comparison could also be more honest. For example, while it is true that the mean $CO_2$ estimate is within 10% of O'Shea, the uncertainties are large. It would be better to state that the numbers agree to within combined uncertainties, which is something like 25 – 30%. This is also true for the abstract.

L420: This should not be surprising given that the observations were in the afternoon, when the diel cycle of ozone typically peaks.

L432: The discussion from here to the end of the section could benefit from improved organization. In particular, I recommend organizing paragraphs and order-of-presentation by plume rather than by chemical species. All paragraphs should have topic sentences. And, it might be clearest if the most likely source(s) are stated at the beginning, followed by evidence to support that identification.

L535: what is meant by "methods that can provide improved quantification of surface interaction"? Fluxes? Please be more specific.

Technical Comments

The word "spike" is used throughout the manuscript to refer to features in the observed time series. In typical usage, this word refers to artifacts (e.g. due to electrical noise). Recommend replacing these words with "features" or "enhancements" or similar.

L219: "that WAS"

L245: "Fig. 4"

L294: which segments are the "background" ones?

L322: "southwesterly wind"

L403: replace "the profile of VOCs by altitude" with "the vertical profile of VOCs."

L418: please quantify "far higher"

L506: which numbered section is this in the time series/map?

L554: replace "known" with "assumed"

L556: "These three flights demonstrate"

L556: This data is not "remote sensing"

Figures in general: text is very hard to read in many cases. Too small. please fix.

Figure 2: line colors for GPH and coastlines are identical. Please change one.

Fig. 4: please mark London with a symbol. Also, do we need all 4 trajectories to get the point across?

Fig. 8 & 11: what is the triangle?

Fig. 9 & 12: Please flip so altitude is on the y-axis. Also, is the "total VOC" bar really that useful a metric?

Fig 14 & 15: Not sure we need both plots; just 15 would suffice. Also, blue color bar clashes with background.

---

## Referee Comment (RC3) · Anonymous Referee #3 · 12 Dec 2019

This manuscript describes measurements around the city of London with an aircraft equipped with various instruments measuring both gases and particles. The data is clearly of excellent quality, and the measurements were well done. The paper has some utility to the measurement and possibly the regional modelling community. The topic is certainly relevant to ACP, as urban outflow and regional impacts of urban outflow is an important issue in atmospheric science. However, ultimately, the entire manuscript relies upon only three flights and limited data. While many publications have used a limited number of flights (sometimes even just 1 flight), this manuscript is severely lacking focus. As such, as currently written, I cannot recommend this be published in ACP, for the reasons noted below. Substantial reorganization and rewriting would be required, although perhaps is possible.

[Figure]

Generally, the manuscript lacks focus. The introduction should clearly set out what is unique or novel about this study, but it does not do so. It is imperative that ones describes in the introduction, how this study is any different than others, and what additional information is gained here. This is especially important here, since there have in fact been other similar studies aboard aircraft around the London area. Without this introductory information, this paper seems like a simple reporting of obtained data without a clear motivation or scientific objective. Determining the relative importance of London outflow is not a sufficient objective and lacks detail, neither is the sampling of an urban plume, as that has been done many times. I suspect the paper would be more coherent if the objectives were clearly stated from the outset.

In the methods section, more information regarding where the flight took place relative to the urban city is needed. How far from the city were you? How far downwind from London were the flights? Were multiple altitudes flown for each flight? A sense of the photochemical age for the air masses should be provide up front, as should a description of the purpose of fling the type of flight conducted.

Only a single flight is not sufficient to say anything meaningful regarding the flux of CO, CO2, CH4. In addition, the method by which the flux was determined was very poorly described. Nothing is mentioned about how the flux below the lowest flight track is determined. This can be a substantial amount but is unclear how this is treated here. Was an extrapolation performed to the ground? This is in fact critical for ground based sources, as the highest concentration of pollutants is often below the lowest flight track, and without this information it is unclear how accurate the estimate would be. Significantly more description of the flux approach is needed.

Regardless, it is unclear how the flux from one flight is illustrative of anything. Nor is it possible to make a meaningful comparison to anything else, due to hourly/daily variability of emissions. Furthermore, some effort into determining the impact of the background subtraction on the flux is required. Finally, this is not technically a "mass balance" approach as stated, as the authors have not gone through the task of determining if a mass balance is actually achieved, particularly through the top of the cylinder.

The paper is generally poorly organized, which makes it very difficult to read. The sections should more likely be organized by scientific objective rather than by flight. However, without clearly stated objectives in this paper, that is a difficult task. Clearly stating the objectives at the beginning of the paper would help to determine how to better organize the rest of the paper.

What is the point of having a section on marine emissions? This section seems to come out of nowhere, and is of minimal value. How did this suddenly become a marine vessel paper? I suggest removing this section unless it fits with the objectives of this paper as a whole. As written it currently does not.

There are far too many figures in this paper to be readable. It reads as a set of observations associated with these figure with no clear outcome. Many of these figures can be in the SI, keeping only the ones that provide evidence of the objective.

Specific items:

Line 82: "These observations match those of the EM25 campaign". If this statement is true, then what is the purpose of this paper?

Line 121: "Local" vs London outflow need to be put in context and properly defined. Since at this point the reader has no idea how far from London the flights were conducted, local and London could be the same thing. If you were flying around London, then presumably everything is "local" to London.

Line 181: it is not clear what is meant by "temporal stability of total aerosol"

Line 529: what is the importance of this statement? It is not clear how this is a "conclusion".

Line 557-559: There is nothing new about this statement. It is quite obvious that "the

factors that control the air pollution buildup in the London area are various and multiple: local emissions, transport from distant sources, terrestrial and marine emissions". This is not a significantly new conclusion here.

---

## Author Comment (AC1) · 2 Mar 2020

We thanks all three anonymous reviewers for their time and feedback. We are confident that the revised manuscript now addresses all of their concerns and we agree that it is a much stronger article as a result.

Reviewer #1:

We thank Reviewer #1 for highlighting the excellence and rigour of our dataset, the quality of our analysis and presentation and the interest of our conclusions to the atmospheric chemistry community.

Specific comments:

1/ The paper compares CH4, CO2 and CO fluxes derived using mass balancing with those reported using highly analogous aircraft sampling and methods by O'Shea et al., 2014 and Pitt et al., 2019. The paper currently discusses the relative quantitative differences with those studies. A further useful dimension to the discussion surrounding that comparison could be more thought as to "why" they differ... I don't believe that difference in method and footprint solely explain the different fluxes, which is the message that is perhaps currently conveyed.

The reviewer makes a good point that the text places undue emphasis on methodological discrepancies and makes very little mention of the temporal differences between flights. We have added text around differences in emissions which provide better balance in the discussion of results (Section 3.3.1) and again in the conclusion. "When considering these data, one should be mindful that aircraft measurements are representative of a single point in time and therefore cannot be aggregated over longer periods. As such they are highly sensitive to meteorology and hence emissions footprint and source strength at the time of measurement. Due to the short duration of and significant separation in time between our and O'Shea's flights, variation in emissions from London (either diurnally, seasonally or longer-term) are likely to be substantial and should be borne in mind when comparing between studies although differences in methodology need also be considered."

2/ The paper often refer to "spikes" or "spiking". I think words such as "concentration enhancement" or "transient enhancement" could be more intuitive.

We have replaced the word "spike" with local / transient enhancement / elevation throughout the manuscript.

3/ All concentrations are reported in units of ppv (e.g. ppbv) in the paper. As far as I understand, most in situ instruments on the FAAM aircraft report molar mass concentration

We can confirm that final datasets released by FAAM and made available via Centre

for Environmental Data Analysis (CEDA) report molar (equivalent to vol vol-1) as used throughout the manuscript rather than mass concentrations.

Technical comments:

1/ Some units do not have a space between quantity and unit, e.g. 450m (and elsewhere where m are used).

This inconsistency has been rectified.

2/ Figure 3 – does not appear well on my screen. Can the axes lines be thickened?

The lines have been thickened, the font size increased and the resolution increased. Note that this is now Figure 2.

3/ Legend on Figure 12 – hard to read – can a larger font size be used?

The font size for the legend, axis titles and labels have been increased.

Reviewer #2:

This paper is largely a description of the observations, with some analysis of emissions that carries large caveats and uncertainties. These particular measurements have not been presented before, but there have been previous similar observations. It is not clear to the reviewer what new or novel things we have learned, or can learn, from this dataset. Some reorganization would help, as would focusing more on current open questions.

We believe that the reorganisation of our analysis and the highlighting of specific novelties in our data set fully addresses the reviewer's concerns and considerably strengthens the resulting manuscript. In particular we have focused on the relative contribution of local sources rather than the influence of London outflow and highlighted the potential of using aircraft measurements to understand air pollution in this complex region.

Specific Comments

Use of WAS data for source identification: there are many examples in the literature of the use of specific hydrocarbon ratios to identify distinct emission sources. Why was this not done here?

This technique is most powerful in situations with few large emissions sources or when tracking a single plume over time and space. We have made it clearer in the text around the analysis of the WAS samples that this could only be done in a limited way here as we were sampling in a region of multiple pollution sources mixing into a relatively regional distributed air mass over very different temporal and spatial scales. However what we have been able to do is advise that emission profiles are changing very substantially over time and some of these traditional methods (particularly benzene:toluene ratios) are likely no longer reliable for positive attribution or estimating air mass age.

Flight segments: It is helpful to number flight segments when comparing the map (Fig. 1) and time series (Figs 5 – 7). However, there are so many flight segments that the map looks cluttered and it is sometimes difficult to identify features discussed in the text. Rather than using one for every 5 minutes, how about 4 – 5 sections per flight?

We tried reducing the number of flight segments (to 6-7 per flight) and while this did make Fig. 1 easier to read it was far harder to pinpoint the individual pollution events we now describe in Section 3.3. We have therefore kept the original versions of these figures.

L342 onward: discussion seems to indicate that the flux estimates shown in Table 3 are not comparable because of differences in the methodology and data used to do the calculation. Is this really the case, or is it just that the footprint is different? Also this makes Table 3 itself questionable, as it is comparing numbers that are not comparable.

The text has been modified in this section as Reviewer #1 raised the same point (see previous response).

Statements regarding the comparison could also be more honest. For example, while it is true that the mean CO2 estimate is within 10% of O'Shea, the uncertainties are large. It would be better to state that the numbers agree to within combined uncertainties, which is something like 25 – 30%. This is also true for the abstract.

We have amended the text accordingly

L420: This should not be surprising given that the observations were in the afternoon, when the diel cycle of ozone typically peaks.

We agree and make this clearer in the text: "slightly higher ($\sim$44 vs. 40 ppbv) in the afternoon than morning, as expected for a secondary pollutant formed as a product of the photochemistry"

L432: The discussion from here to the end of the section could benefit from improved organization. In particular, I recommend organizing paragraphs and order-of-presentation by plume rather than by chemical species. All paragraphs should have topic sentences. And, it might be clearest if the most likely source(s) are stated at the beginning, followed by evidence to support that identification.

This suggestion has been followed, with 4 specific plumes described in results Section 3.3 with additional headings as required and attribution where possible. We thank the reviewer for this as it has indeed improved the flow of the discussion.

L535: what is meant by "methods that can provide improved quantification of surface interaction"? Fluxes? Please be more specific.

This line has been removed in the reorganisation and refocusing of the manuscript

Technical Comments

The word "spike" is used throughout the manuscript to refer to features in the observed time series. In typical usage, this word refers to artifacts (e.g. due to electrical noise). Recommend replacing these words with "features" or "enhancements" or similar. We

have replaced the word "spike" with local / transient enhancement / elevation throughout the manuscript.

L219: "that WAS"

The sentence makes sense as it stands

L245: "Fig. 4"

Amended, thank you

L294: which segments are the "background" ones?

The text has been substantially altered throughout the results section. However, segment numbers have now been added each time a feature is referred to.

L322: "southwesterly wind"

Amended, thank you

L403: replace "the profile of VOCs by altitude" with "the vertical profile of VOCs."

Text amended accordingly.

L418: please quantify "far higher"

The following text has been added: "(peaking at 0.05 ppbv vs. <0.01 ppbv)"

L506: which numbered section is this in the time series/map?

The text has been substantially altered throughout the results section. However, segment numbers have now been added throughout.

L554: replace "known" with "assumed"

Text amended accordingly

L556: "These three flights demonstrate"

Text amended accordingly.

L556: This data is not "remote sensing"

Some parts of the community consider anything above tower-based measurements to be "remote sensing". However we have re-phrased this sentence to simply read "airborne measurements".

Figures in general: text is very hard to read in many cases. Too small. please fix.

Text size increased

Figure 2: line colors for GPH and coastlines are identical. Please change one.

Figure 2 has in fact been removed from the final version

Fig. 4: please mark London with a symbol. Also, do we need all 4 trajectories to get the point across?

Now Fig. 3. A symbol marking central London has been added to this and the other back-trajectory plots (Figs. 7 & 11). We believe we do need all four trajectories as an important requirement for the flux estimation method is that the plume has clear and distinct edges which these demonstrate.

Fig. 8 & 11: what is the triangle?

Now Fig. 7 & 11. This is already stated in the caption of both figures: "the location of FAAM BAe-146 (black triangles)"

Fig. 9 & 12: Please flip so altitude is on the y-axis. Also, is the "total VOC" bar really that useful a metric?

Now Fig. 10 & 12. Altitude now on y-axis as requested and the total VOC bar has been removed

Fig 14 & 15: Not sure we need both plots; just 15 would suffice. Also, blue color bar clashes with background.
* * *
Interactive
comment

We have removed Fig. 15 as Fig. 14 provided an overview of all shipping. The colour bar has been altered. (Note that Fig. 14 is now Fig. 9)

Reviewer #3

Generally, the manuscript lacks focus. The introduction should clearly set out what is unique or novel about this study, but it does not do so. It is imperative that ones describes in the introduction, how this study is any different than others, and what additional information is gained here. This is especially important here, since there have in fact been other similar studies aboard aircraft around the London area. Without this introductory information, this paper seems like a simple reporting of obtained data without a clear motivation or scientific objective. Determining the relative importance of London outflow is not a sufficient objective and lacks detail, neither is the sampling of an urban plume, as that has been done many times. I suspect the paper would be more coherent if the objectives were clearly stated from the outset.

The paper has now been substantially reorganised in response to the comments of both this reviewer and Reviewer 2. The objectives are now stated clearly at the beginning and the main aim identified as assessing the extent to which local sources play an important role in influencing air pollution episodes in a region proximal to a megacity. We believe that the paper is substantially more coherent as a result and thank both reviewers for their suggestions.

In the methods section, more information regarding where the flight took place relative to the urban city is needed. How far from the city were you? How far downwind from London were the flights? Were multiple altitudes flown for each flight? A sense of the photochemical age for the air masses should be provide up front, as should a description of the purpose of fling the type of flight conducted.

The distance from London has been added and the different altitudes of the flight legs, already shown in the accompanying figures and in the results section, have been stated

[Figure]

Only a single flight is not sufficient to say anything meaningful regarding the flux of CO, CO2, CH4. In addition, the method by which the flux was determined was very poorly described. Nothing is mentioned about how the flux below the lowest flight track is determined. This can be a substantial amount but is unclear how this is treated here. Was an extrapolation performed to the ground? This is in fact critical for ground based sources, as the highest concentration of pollutants is often below the lowest flight track, and without this information it is unclear how accurate the estimate would be. Significantly more description of the flux approach is needed.

The description of the flux calculation has been improved and the interpolation and extrapolation methods and assumptions stated more clearly.

Regardless, it is unclear how the flux from one flight is illustrative of anything. Nor is it possible to make a meaningful comparison to anything else, due to hourly/daily variability of emissions. Furthermore, some effort into determining the impact of the background subtraction on the flux is required.

The text around the validity of the comparison has been amended to highlight the uncertainties due in particular to the time variation in emission sources. The paper that we have made our comparison against was also only able to use the mass balance approach in one flight as the conditions required for it to be appropriate are only rarely encountered. we therefore highlight that it offers the potential to constrain bottom-up estimates but is limited by spatial and temporal resolution and coverage.

The impact of the background subtraction has been included.

Finally, this is not technically a "mass balance" approach as stated, as the authors have not gone through the task of determining if a mass balance is actually achieved, particularly through the top of the cylinder.

The term "mass balance" has been removed from throughout the manuscript. Instead we refer to calculating emission fluxes. We have also added a statement regarding the

very clear cap at the top of the cylinder based on the boundary layer conditions (shown in Fig 2) and the concentrations observed at different altitudes during that flight. The NOx profile (Fig R1 attachment but not included in the paper) shows a clear upper altitude bound.

The following text has been added to this section of the manuscript: "We assumed that air below the lowest flight track was well mixed and that the full vertical profile of the plume was captured by these flight legs. Boundary layer height was estimated from temperature-humidity profiles to be between 800 and 1000 m while the plume was being sampled. Vertical profiles for NOx only showed significant enhancement below these heights indicating a lack of mixing into the free troposphere. NOx was used to indicate this, as its shorter lifetime leads to near zero concentrations above the boundary layer, whereas the difference is less pronounced in the longer-lived CO/CO2/CH4. We assumed that air below the lowest flight track was well mixed and that the full vertical profile of the plume was captured by these flight legs. Boundary layer height was estimated from temperature-humidity profiles to be between 800 and 1000 m while the plume was being sampled. Vertical profiles for NOx only showed significant enhancement below these heights." indicating a lack of mixing into the free troposphere. NOx was used to indicate this, as its shorter lifetime leads to near zero concentrations above the boundary layer, whereas the difference is less pronounced in the longer-lived CO/CO2/CH4.

The paper is generally poorly organized, which makes it very difficult to read. The sections should more likely be organized by scientific objective rather than by flight. However, without clearly stated objectives in this paper, that is a difficult task. Clearly stating the objectives at the beginning of the paper would help to determine how to better organize the rest of the paper.

The paper has now been substantially reorganised in response to the comments of both this reviewer and Reviewer 2. The objectives are now stated clearly at the beginning and the main aim identified as assessing the extent to which local sources play an

important role in influencing air pollution episodes in a region proximal to a megacity.

What is the point of having a section on marine emissions? This section seems to come out of nowhere, and is of minimal value. How did this suddenly become a marine vessel paper? I suggest removing this section unless it fits with the objectives of this paper as a whole. As written it currently does not.

It has been re-written and now does fit the stated objectives which were to identify local sources that were substantial enough to be clearly visible as pollution episodes even in a region dominated by megacity outflow.

There are far too many figures in this paper to be readable. It reads as a set of observations associated with these figure with no clear outcome. Many of these figures can be in the SI, keeping only the ones that provide evidence of the objective.

One table and 3 figures have been removed and with the new framing of the paper the remaining ones are used to provide evidence / support for our stated objectives.

Specific items:

Line 82: "These observations match those of the EM25 campaign". If this statement is true, then what is the purpose of this paper?

Line 82 is referring to the Aruffo observations not ours. This has been made clearer.

Line 121: "Local" vs London outflow need to be put in context and properly defined. Since at this point the reader has no idea how far from London the flights were conducted, local and London could be the same thing. If you were flying around London, then presumably everything is "local" to London.

We have stated the distance from central London and made our distinction of London vs local clearer in the newly reorganised manuscript.

Line 181: it is not clear what is meant by "temporal stability of total aerosol"

This and the following sentence have now been re-worded to read: "Concentration of ultrafine aerosol was monitored using a condensation particle counter (CPC; Model 3786, TSI Incorporated, MN, USA) at 1 Hz, while an additional optical particle counter (OPC; Grimm Aerosol Technik GmbH & Co. KG, Germany) was used to correctly count and size aerosol particles (Allen et al., 2011)."

Line 529: what is the importance of this statement? It is not clear how this is a "conclusion". It's a summary preamble to the conclusions. Its importance is that it enabled us to sample a range of different pollution events over the course of the two days.

Line 557-559: There is nothing new about this statement. It is quite obvious that "the factors that control the air pollution build up in the London area are various and multiple: local emissions, transport from distant sources, terrestrial and marine emissions". This is not a significantly new conclusion here.

This has been re-phrased and tied more clearly to our stated objectives: "These three flights demonstrate the power of airborne measurements which can be used for targeted sorties to provide direct source attribution (or test hypotheses of sources) and for longitudinal studies over time to provide evidence of new or changing emission sources or source profiles to inform and constrain bottom-up emissions inventories. They also provide clear evidence that relatively small local sources can still play a significant role in air pollution in a megacity region, particularly downwind where they exacerbate high "background" levels of pollution. The factors that control the buildup of air pollution in the London area are various and multiple: local emissions, transport from distant sources, terrestrial and marine emissions. In the highly complex environment around a megacity where a high background level of pollution mixes with a variety of local sources at a range of spatial and temporal scales, the use of unvarying VOC:VOC ratios may not be valid given the different ages of the air. It is necessary to consider and constrain all of the contributing factors to understand the problem and to develop effective mitigation and control strategies."

**Fig. 1.** Figure R1. Vertical profile of NOx during flight C016

---

## Author Response (AR2)

**Editor Decision: Publish subject to minor revisions (review by editor)**

We thank the Editor (Drew Gentner) for taking the time to read our response and revised manuscript so thoroughly at what we appreciate is a difficult time.

(1) There are a wide number of sources of light alkanes that could contribute to enhancements downwind of an urban area. If you are going to attribute the enhancements as "likely" specifically from petroleum processing-related facilities, it would be necessary to clearly point to where the enhancements were high in WAS samples with back trajectories to a site AND show how the relative ratios of light alkanes from these sources match the chemical source profile expected for petroleum operations (which are varied depending on facility type and feedstock composition). I worry your conclusion "Ethane, propane, n-butane and cyclopentane, exhibit the highest average concentrations across all three flights and can likely be similarly attributed to petrochemical refining and natural gas processing", which also appears in the results section is attributing the enhancement entirely to that source sector (since past studies had a similar conclusion). These efforts should be taken to substantiate this claim, or you can restructure your conclusions (in both locations) as a suggestion or hypothesis.

As our flights traversed areas downwind of numerous point sources VOCs, we found concentrations of alkanes were consistently high across much of the region. We accept the point that in the case of the light alkanes we are unable to attribute the enhancements to a specific source or process. We have re-phrased our results section to make that clear and removed a definite attribution from our conclusions.

Lines 445-447 now read: "Petrochemical refining and natural gas processing have previously been identified as strong sources of ethane, propane and n-butane. This may explain the enhancements here as there are several large processing facilities east and north-east of London but given the wide distribution of these high concentrations it was not possible to identify the precise source."

Lines 569-572 now read: "Ethane, propane, n-butane and cyclopentane made up the highest proportion of VOCs across all three flights. While there are a considerable number of petrochemical refining and natural gas processing facilities around London, the presence of these VOCs was too ubiquitous for us to be able to unambiguously determine the source."

(2) This conclusion is unclear to me "They also provide clear evidence that relatively small local sources can still play a significant role in air pollution in a megacity region." I think I understand the point you are trying to make, but the statement formulation of "relatively small" and "significant role" are contradictory as stated. Are they small in footprint? Source activity? Right now it reads as if the emissions are small.

We have clarified this statement to read: "They also provide clear evidence that even in a region where background pollution concentrations are dominated by emissions from a megacity, relatively small point sources can still play a significant role in local air pollution, particularly downwind where they exacerbate already high levels."

(3) It would seem to me like the term "sorties" (appears twice) is not most appropriate here, and is likely to confuse readers.

We have replaced "sorties" with "techniques" and "campaigns"

(4) "CPC is often used to indicate particle number concentrations in figures and tables, e.g. "CPC Counting" in Figure 8. Please call this particle number concentration (Similar for panel titles and axis titles in Figures 4-6) and

Here are 2 examples (that would not be caught by copy-editors):
a. Figures 10 and 12 list "npentane" in the legend, as opposed to "n-pentane"
b. Both those captions state that altitude is on the x-axis, it is in fact on the y-axis

**Legends and captions of Figs 10 and 12 changed accordingly.**

[revised manuscript text omitted]

**C017**

| Run | Ave. alt. (m) | [CO] (ppbv) | [CH$_4$] (ppbv) | [NO$_x$] (ppbv) | [O$_3$] (ppbv) | PM (cm$^{-3}$) | [CO] (ppbv) | [CH$_4$] (ppmv) | [NO$_x$] (ppbv) | [O$_3$] (ppbv) | PM (cm$^{-3}$) | [CO] (ppbv) | [CH$_4$] (ppmv) | [NO$_x$] (ppbv) | [O$_3$] (ppbv) | PM (cm$^{-3}$) |
|---|---|---|---|---|---|---|---|---|---|---|---|---|---|---|---|---|
| | | **Full flight leg** | | | | | **"West plume" (52.57N, 1.00E)** | | | | | **"East plume" (52.42N, 1.45E)** | | | | |
| 4 | 263 | 95.59 | 2.085 | 1.509 | 39.39 | 7415 | 96.68 (101.6) | 2.082 (2.103) | 1.324 (2.080) | 39.08 (40.86) | 8517 (13800) | 94.17 (102.2) | 2.082 (2.094) | 2.411 (4.855) | 39.36 (41.85) | 5874 (6660) |
| 5 | 408 | 97.04 | 2.084 | 1.401 | 40.27 | 6969 | 99.48 (105.6) | 2.089 (2.100) | 1.520 (2.078) | 39.67 (40.94) | 8083 (9650) | 94.38 (99.48) | 2.079 (2.092) | 1.954 (4.476) | 40.69 (43.54) | 5754 (10300) |
| 6 | 522 | 97.29 | 2.083 | 1.519 | 40.09 | 6664 | 100.1 (107.2) | 2.090 (2.097) | 1.579 (2.233) | 39.82 (40.99) | 7851 (12700) | 94.90 (101.1) | 2.078 (2.091) | 2.013 (2.908) | 40.09 (43.45) | 5760 (8090) |
| 7 | 674 | 97.85 | 2.083 | 1.415 | 40.23 | 5936 | 100.5 (108.5) | 2.087 (2.097) | 1.704 (3.187) | 39.39 (41.46) | 6269 (8570) | 95.15 (101.3) | 2.076 (2.086) | 1.391 (3.182) | 40.84 (42.75) | 5763 (23100) |
| 8 | 831 | 97.13 | 2.079 | 1.277 | 41.04 | 5682 | 100.7 (106.3) | 2.083 (2.092) | 1.804 (3.063) | 39.88 (42.32) | 5581 (7000) | 95.14 (99.06) | 2.073 (2.085) | 1.291 (1.928) | 42.40 (44.01) | 5915 (7020) |

**C018**

| Run | Ave. alt. (m) | [CO] (ppbv) | [CH$_4$] (ppbv) | [NO$_x$] (ppbv) | [O$_3$] (ppbv) | PM (cm$^{-3}$) | [CO] (ppbv) | [CH$_4$] (ppbv) | [NO$_x$] (ppbv) | [O$_3$] (ppbv) | PM (cm$^{-3}$) | [CO] (ppbv) | [CH$_4$] (ppbv) | [NO$_x$] (ppbv) | [O$_3$] (ppbv) | PM (cm$^{-3}$) |
|---|---|---|---|---|---|---|---|---|---|---|---|---|---|---|---|---|
| | | **Full flight leg** | | | | | **"West plume" (52.57N, 1.00E)** | | | | | **"East plume" (52.46N, 1.30E)** | | | | |
| 5 | 287 | 105.1 | 2.085 | 2.785 | 41.52 | 5754 | 111.9 (128.1) | 2.102 (2.116) | 2.498 (5.686) | 43.27 (45.76) | 6777 (19300) | 97.92 (107.7) | 2.067 (2.080) | 4.157 (7.760) | 38.50 (47.32) | 5222 (6450) |
| 6 | 415 | 105.2 | 2.084 | 2.414 | 43.03 | 5460 | 112.5 (126.0) | 2.101 (2.116) | 2.624 (4.919) | 43.46 (47.88) | 5984 (9480) | 97.27 (104.3) | 2.067 (2.079) | 2.909 (4.177) | 41.09 (47.38) | 4949 (6150) |
| 7 | 533 | 101.8 | 2.077 | 2.150 | 44.09 | 5388 | 116.2 (123.8) | 2.097 (2.116) | 2.440 (4.174) | 42.47 (45.54) | 5741 (8420) | 99.09 (115.6) | 2.067 (2.082) | 2.697 (3.780) | 42.20 (49.48) | 5247 (6370) |
| 8 | 686 | 103.6 | 2.079 | 1.993 | 43.96 | 5167 | 110.5 (127.7) | 2.093 (2.114) | 2.404 (4.066) | 43.09 (47.82) | 5476 (8390) | 98.66 (104.5) | 2.068 (2.074) | 2.172 (4.919) | 43.28 (47.51) | 5159 (6010) |
| 9 | 843 | 103.8 | 2.081 | 1.835 | 44.11 | 5077 | 107.6 (122.0) | 2.087 (2.112) | 1.985 (3.550) | 41.61 (45.51) | 4821 (7370) | 99.12 (104.5) | 2.071 (2.080) | 1.830 (2.780) | 46.12 (49.47) | 5411 (7000) |
| 10 | 998 | 103.4 | 2.078 | 1.718 | 44.08 | 5103 | 108.7 (125.0) | 2.086 (2.112) | 2.311 (4.409) | 41.83 (46.14) | 4758 (6440) | 101.2 (107.8) | 2.076 (2.082) | 1.441 (1.774) | 48.74 (50.57) | 5591 (6430) |
| 11 | 1155 | 102.6 | 2.075 | 1.544 | 41.29 | 5305 | 103.4 (114.9) | 2.075 (2.100) | 1.733 (2.809) | 40.12 (42.48) | 3914 (5110) | 104.8 (113.9) | 2.082 (2.094) | 1.677 (2.619) | 47.96 (49.30) | 5193 (6440) |

**Table 3: Overview of**  **CO, CH$_4$, NO$_x$ and O$_3$**  **and** **PM (particle** number  **concentration** **measured during the reciprocal runs on flights C017 (top) and C018 (bottom). The full flight leg values are averages along each run, the West plume are average and maximum (in parentheses) concentrations within the West plume and the same for the East plume. The average altitude is shown for each leg; the average latitude and longitude of the centre of each plume are shown** **.**